# Genetic Impact of HOTAIR, LINC00951, POLR2E and HULC Polymorphisms in Histopathological and Laboratory Prognostic Factors in Esophageal Cancer in the West: A Case-Control Study

**DOI:** 10.3390/cancers16030537

**Published:** 2024-01-26

**Authors:** Efstratia Baili, Maria Gazouli, Andreas C. Lazaris, Prodromos Kanavidis, Maria Boura, Adamantios Michalinos, Alexandros Charalabopoulos, Theodore Liakakos, Andreas Alexandrou

**Affiliations:** 1Upper Gastrointestinal and General Surgery Unit, First Department of Surgery, Laiko General Hospital, School of Medicine, National and Kapodistrian University of Athens, 15772 Athens, Greece; pkanav@med.uoa.gr (P.K.); m.mpoura@hotmail.com (M.B.); alexcharala@med.uoa.gr (A.C.); theodlia@med.uoa.gr (T.L.); andreasalex@med.uoa.gr (A.A.); 2King’s Health Partners, London SE1 9RT, UK; 3Laboratory of Biology, School of Medicine, National and Kapodistrian University of Athens, 15772 Athens, Greece; mgazouli@med.uoa.gr; 4First Department of Pathology, School of Medicine, National and Kapodistrian University of Athens, 15772 Athens, Greece; alazaris@med.uoa.gr; 5School of Medicine, European University of Cyprus, Nicosia 1516, Cyprus; a.michalinos@euc.ac.cy

**Keywords:** Single-Nucleotide-Polymorphisms (SNPs), esophageal cancer, esophagogastric junction carcinoma, lncRNAs, HOTAIR rs920778, LINC00951 rs11752942, POLR2E rs3787016, HULC rs7763881

## Abstract

**Simple Summary:**

Single-Nucleotide-Polymorphisms in long non-coding RNAs are correlated with esophageal carcinogenesis, yet research remains restricted in Asian ethnicities. Herein, FFPE specimens were obtained from surgically treated esophageal cancer patients for genotyping deriving from European ancestry. HULC rs7763881 polymorphism was not associated with cancer predisposition. LINC00951 rs11752942 was underrepresented in Ca19.9 elevated subgroup. HOTAIR rs920778 was more frequent in both worse differentiation grade and positive Signet Ring Cell and Ca19.9 subgroups. POLR2E rs3787016 was identified less frequently in Adenocarcinoma, Signet Ring Cell, and Diffuse histological subtypes, as well as in Perineural, Lymphovascular, and Perivascular Invasion positive, while it was found more often in CEA positive subgroup of the whole cancer cohort. Taken together, these lncRNAs polymorphisms are promising not only as future therapeutic agents but also as novel markers for predictive analysis of esophageal cancer risk and oncological outcomes including survival.

**Abstract:**

Long non-coding RNAs’ HOTAIR rs920778, LINC00951 rs11752942, POLR2E rs3787016, and HULC rs7763881 are progressively reported having a close genetic affinity with esophageal carcinogenesis in the East. Nonetheless, their correlation with variables already endorsed as significant prognostic factors in terms of staging, guiding treatment and predicting recurrence, metastasis, and survival have yet to be explored. Herein, we investigated their prognostic value by correlating them with clinicopathological and laboratory prognostic markers in esophageal cancer in the West. Formalin-fixed paraffin-embedded tissue specimens from 95 consecutive patients operated on for esophageal cancer between 2014 and 2018 were compared with 121 healthy community controls. HULC was not detected differently in any of the cancer prognostic subgroups. LINC00951 was underrepresented in Ca19.9 elevated subgroup. HOTAIR was more frequent in both worse differentiation grade and positive Signet-Ring-Cell and Ca19.9 subgroups. POLR2E was identified less frequently in Adenocarcinoma, Signet-Ring-Cell, and Diffuse histologies, as well as in Perineural, Lymphovascular, and Perivascular Invasion positive, while it was overrepresented in CEA positive subgroup. These lncRNAs polymorphisms may hold great potential not only as future therapeutic agents but also as novel markers for predictive analysis of esophageal cancer risk, clinical outcome, and survival. Clinical implications of these findings need to be validated with prospective larger sample-size studies.

## 1. Introduction

Molecular biology and epigenetics are currently in the spotlight in the investigation of esophageal and esophagogastric junction oncogenesis [1]. Esophageal cancer (EC) is still the seventh most common cancer worldwide with estimated 604,100 new patients in 2020, ranking sixth in overall mortality accounting for 544,076 new deaths as per GLOBOCAN 2021 [2]. Despite the so-far notable advances achieved in earlier diagnosis and multimodal treatment, the prognosis for EC remains poor, with a 5-year survival rate of 19% [3]. Tumor recurrence, metastasis, and resistance to chemoradiotherapy are major contributing factors to poor survival outcomes [4,5].

The development of EC is a multifactorial process and comes as a consequence of not only environmental and genetic factors but also of specific tumor behavior characteristics, which may vary among EC patients. In an effort to better predict disease trajectory overtime, scientific communities such as the American Joint Committee on Cancer (AJCC) and the Union for International Cancer Control (UICC) have incorporated certain prognostic factors to aid the staging efforts, and thereby the risk-assessment process, on disease recurrence and metastasis and ultimately the estimates on survival [6].

Histopathologic cell type is the cornerstone in the staging and risk-assessment process in EC. The two most common histologic subtypes, esophageal squamous cell carcinoma (ESCC) and esophageal adenocarcinoma (EAC), vary substantially in terms of aetiopathogenesis, genetic susceptibility, clinical features, and prognosis, as well as gender and geographic distribution. Approximately half of EC cases are EAC in Europe, Oceania, and some Western countries, including the United States [7], whereas ESCC remains the dominant type in other areas of the world, particularly in Asia and Africa. A male predominance is observed worldwide in EC and Gastric Cancer (GC), with male-to-female ratios of 6.7:1 for EAC, 3.3:1 for ESCC, and 4:1 for GC [8]. While ESCC is essentially in decline, EAC incidence rates have escalated rapidly over the past decades [9]. In addition to histology type [10], other prognostic factors of major significance in predicting disease evolution include histologic grade of differentiation (G status) for EAC, tumor location and length for ESCC, infiltration potential in terms of Perineural Invasion (PNI), Lymphovascular Invasion (LVI) and Perivascular Invasion (PVI) Status, presence or absence of Signet Ring Cell (SRC), and Intestinal or Diffuse histological subtype in gastric adenocarcinoma, as per Lauren Classification [11].

Along with these prognostic factors, which are directly related to final histopathologic characteristics of the primary tumor, preoperative serum tumor markers in the form of Carcinoembryonic Antigen (CEA) and Carbohydrate antigen 19.9 (Ca19.9) have also been proven useful in diagnosis, guiding management, and predicting response to treatment and survival in EC [12].

Long non-coding RNAs (LncRNAs) are transcription products longer than 200 nucleotides that do not participate in protein expression but regulate gene expression at epigenetic, transcriptional, and post-transcriptional levels, thus influencing processes such as cell growth, apoptosis, and protein activity regulation [13]. Aberrant expression of lncRNAs is associated with cellular malignant potential [14]. When studied for colorectal adenocarcinoma, lncRNAs were found to be related with tumor size, histological subtypes, differentiation grade, Dukes staging, lymph node (LN) involvement, distant metastasis, disease-free survival (DFS), and overall survival (OS) [15]. POLR2E was detected among gene expression profile pathways in bladder cancer patients in Egypt [16]. Associations between EC and the most common genetic variants as lncRNA Single-Nucleotide-Polymorphisms (SNPs) have been detected in Asian studies [17,18]. Abnormal expression of HOTAIR in digestive cancers has recently been correlated with histopathological variables such as G status, indicating that HOTAIR may act as a prognostic biomarker to predict survival in various types of cancers such as GC [19] and ESCC [20]. Zhang et al. [21] investigated the clinical role of HOTAIR expression as a prognostic indicator in digestive cancers by conducting a meta-analysis in 2018. When the authors compared poor versus well differentiated tumors in a total of five studies encompassing 386 patients, G status was found to be positively associated with high HOTAIR expression (OR: 1.65, *p* = 0.040), suggesting it could serve as a novel prognostic biomarker in patients with digestive cancers.

Nevertheless, the mechanism of lncRNAs and their polymorphisms in esophagogastric cancer susceptibility has only been sporadically studied. Based on previous research, we intended to explore the potential effects of HOTAIR rs920778, LINC00951 rs11752942, POLR2E rs3787016, and HULC rs7763881 in histopathological and laboratory risk factors and thus their prognostic significance in an EC population of European/Greek descent.

## 2. Materials and Methods

### 2.1. Study Design

This tertiary referral hospital-based case-control study was designed according to ‘Strengthening the Reporting of Observational Studies in Epidemiology’ STROBE Guidelines (Appendix A) [22]. The developed research protocol was strictly followed by all participating authors/researchers. Approval was obtained prior to the start from the Institutional Review Board of Laiko General Hospital and Ethics Committee of School of Medicine-National and Kapodistrian University of Athens (NKUA), Greece (IRB no: 18.01.2018/24), including both case and control populations. All performed procedures were consistent with the ethical standards of the Helsinki Declaration 1964 and later versions. All participants were consented prior to enrollment.

The study was conducted over a nine-year period, with the recruitment phase set between 25 March 2014 and 25 September 2018 and the follow-up phase with an end-date of 30 June 2023 for all recruited subjects. Two independent authors (EB, MB) extracted the data retrospectively from our prospectively collected Upper GastroIntestinal (UGI) Cancer database, including data from theatres, surgical/medical records, electronic/paper notes from inpatient and outpatient visits, and investigations performed in both public and private sectors. Discrepancies in data extraction were resolved after consensus with a third independent author (AM).

### 2.2. Patient Selection: Inclusion-Exclusion Criteria

All consecutive adults who underwent surgery for histologically-confirmed malignancy involving the middle-third or lower-third part of the esophagus, or the esophagogastric junction (EGJ) (Siewert I–III) [23] at the Department of UGI Surgery, Laiko General Hospital, School of Medicine—NKUA, Greece, were deemed eligible for inclusion. The clinical and pathological staging, as well as all the included definitions, conform with the AJCC/UICC Guidelines—TNM 8th edition [6]. As such, cancers crossing the EGJ with their epicenter in the proximal 2 cm of the stomach (EGJ-Siewert II) were staged and treated as EC, whereas cancers crossing the EGJ with their epicenter in the proximal 2 to 5 cm of the stomach (EGJ-Siewert III) were staged and treated as GC. All patients were risk-assessed and clinically staged with physical examination, tumor markers, computed tomography, and gastroscopy. At the time of diagnosis, all were evaluated by our dedicated Cancer Multidisciplinary Team, which formulated the appropriate multimodal treatment strategy as per international National Comprehensive Cancer Network (NCCN) [24] and European Society for Medical Oncology (ESMO) [25] guidelines. Exclusion criteria were: (a) non-adults, (b) cancer located on the cervical esophagus, (c) patients submitted to emergency surgery, (d) esophagogastric malignancy family history.

Our control group comprised community subjects recruited from the Department of Molecular Biology, School of Medicine, NKUA, Greece, with no self-reported history of cancer at any site. Both cases and controls were unmatched, of European/Greek ancestry, and resided in the geographical region of Greece.

### 2.3. Data Extraction: Primary-Secondary Variables of Interest

Our study endpoints were to identify potential associations of four lncRNAs’ polymorphisms (HOTAIR rs920778, LINC00951 rs11752942, POLR2E rs3787016, and HULC rs7763881) with histopathological (primary endpoint) and laboratory (secondary endpoint) prognostic markers in esophagogastric cancer in a western population such as Greece. To this end, our histopathological variables of interest encompassed histologic subtypes as EAC and ESCC, grade of differentiation (Gx-3 status), Perineural Invasion (PNI), Lymphovascular Invasion (LVI) and Perivascular Invasion (PVI) Status, presence or absence of Signet-Ring-Cell (SRC), and Intestinal or Diffuse subtypes in EAC subpopulation. Additionally, our laboratory variables of interest encompassed preoperative serum levels of tumor markers in the form of Carcinoembryonic Antigen (CEA) and Carbohydrate antigen 19.9 (CA19.9) in the whole EC cohort. CEA defined as Positive >5 ng/mL, whereas Ca19.9 defined as Positive >37 U/mL.

Further data extracted from our UGI Cancer Database were: (1) demographics such as age, gender, and surgical candidates’ preoperative health as per American Society of Anesthesiologists’ classification (ASA I-V) [26]; (2) primary tumor location, neoadjuvant chemoradiotherapy if offered; (3) date/type of surgical operation, lymphadenectomy extent; (4) final histopathological characteristics as tumor size and location, LN harvest and infiltration, AJCC stage, neoadjuvant treatment effect, proximal, distal (R1–3), and circumferential resection margins (CRM, mm) as per World Health Organization (WHO) and College of American Pathologists (CAP) recommendations [27]; (5) (%) minor and major complications (90-days), in-hospital mortality (90-days), follow-up length (months), adjuvant treatment where applicable, date/type of recurrence, disease-free survival (DFS, months), and overall survival (OS, months). The complications’ severity was based on Clavien–Dindo classification system [28], with minor complications defined as Grade <II and major as Grade >IIIa. Recurrence date was set as the date of the first investigation documenting the recurrence/metastasis. DFS was defined as the period from the surgery date and the first recurrence date. OS was defined as the period between operation date and patient’s death.

### 2.4. Sample Collection and Preparation for Genetic Analysis

The most senior surgeon (TL) supervised all surgical operations performed during the recruitment period. Surgical tissue specimens for all enrolled patients were transferred, after completion of the surgical operation, to the First Department of Pathology, School of Medicine, NKUA, Greece. After gross pathologic examination and marking of the margins, each specimen was formalin fixed and paraffin embedded. Standard fixation methods to preserve nucleic acid integrity were used, including 10% neutral-buffered formalin fixed for 24–72 h. Once paraffin embedded, the tissue samples were then sectioned with a microtome and placed on a glass slide to formulate microscopic slides ready to be microscopically examined by the pathologists. When necessary, the embedding process was reversed to remove the paraffin wax out and allow for staining of the sections, as with Hematoxylin and Eosin (H&E) staining. All tissue samples were reviewed by two independent pathologists. The most senior pathologist (ACL) assessed all the microscopic slides for each specimen and selected the slide and its corresponding FFPE tissue block with the highest tumor burden in preparation for nucleic acid extraction.

### 2.5. Genotyping of HOTAIR rs920778, LINC00951 rs11752942, POLR2E rs3787016 and HULC rs7763881

The nominated FFPE tissue blocks with the highest tumor burden were subsequently transferred to Molecular Biology Laboratory, School of Medicine, NKUA, Greece. The percentage of tumor cells in each sample was minimum 50%. One to two −1 mm diameter punches were sampled from the FFPE blocks. The punches were deparaffinized, homogenized, and proteinase K digested. Next, the genomic DNA/RNA extraction was performed using a commercial RNA Extraction Kit from FFPE Samples (NucleoZOL, Macherey-Nagel, Düren, Germany). LncRNAs genotypes were identified through the “polymerase chain reaction-restriction fragment length polymorphism” (PCR-RFLP) or allele specific PCR (AS-PCR) depending on the SNP.

The following oligonucleotide primers and enzymes were used for PCR-RFLP: For HOTAIR rs920778 (C>T): Forward-5′-TTACAGCTTAAATGTCTGAATGTTCC-3′ and Reverse-5′-GCCTCTGGATCTGAGAAAGAAA-3′ with Restriction endonuclease MspI. For POLR2E rs3787016 (T>C): Forward-5′-CATCAACATCACGCAGCACG-3′ and Reverse-5′-CCCTGTCCTCCAAGCACTCAT-3′ with NLaIII restriction site. The following oligonucleotide primers were used for AS-PCR: For LINC00951 rs11752942 A>G: Forward-5′-GGGGCAAGAAGGTCAATA-3′, Forward-5′-GGGGCAAGAAGGTCAATG-3′ and Reverse-5′-GGGAATCTGCTGGGCT-3′. For HULC rs7763881: Forward-A: 5′-TGTAGTTCCAGTTTGTCTGAA-3′, Forward-C: 5′-TGTAGTTCCAGTTTGTCTGAC-3′ and Reverse: 5′-TGAACAAGTTGGTTGATCTTTAGC-3′.

The most senior molecular biologist (MG) designed and supervised the experiments as per the published methodology in [29,30].

### 2.6. Statistical Analysis

Descriptive statistical analysis was conducted for all the encountered parameters, measuring the accumulated values. All variables are reported as means and medians with their corresponding standard deviations, ranges, and proportions. We assessed the relationship between lncRNAs’ gene polymorphisms and EC and EAC cancer susceptibility by determining the genotype and allele frequencies for all variables of interest in both cases and controls. Genotype frequencies were compared using Fisher’s exact test with Yate’s continuity correction. Odds ratios (ORs) and 95% confidence intervals (95% CI) were calculated using the approximation of Woolf. To summarize the ORs of the four polymorphisms, we applied five genetic models: allele contrast, homozygous, heterogeneous, dominant, and recessive models (AA, homozygotes for the common allele; AB, heterozygotes; BB, homozygotes for the rare allele). Survival analysis was performed by using Cox proportional hazards models both for continuous and categorical variables and log-rank tests for continuous variables. The probability *p*-values were two tailed and *p* < 0.05 was adopted as the statistically significant level. Censoring date was 30 June 2023. When investigated for conformity with the Hardy–Weinberg equilibrium, we observed no significant deviation from expected numbers in all participating cases. Statistical analysis was performed using R, version 4.0.4 (R Project for Statistical Computing).

## 3. Results

### 3.1. Study Population, Clinicopathological, Surgical, Oncological Outcomes and Survival Analysis for the Prognostic Variables of Interest

All enrolled study subjects were adults of European/Greek ancestry. They were divided into two groups, incorporating FFPE tissue samples from N = 95 consecutive esophageal/EGJ cancer patients submitted to surgical treatment as a case-group and blood samples from n = 121 cancer-free community subjects as a control-group. As per AJCC-8th Edition, based on tumor location and clinical staging, N = 95 patients comprising the surgical group were submitted to Ivor Lewis and McKeown esophagectomy for either ESCC (N = 6) or middle/lower EAC or EAC-EGJ Siewert I/II (N = 61). N = 21 patients with EAC-EGJ Siewert III adenocarcinoma underwent total extended gastrectomy. N = 7 EAC patients with small tumors extending marginally between EGJ Siewert II and III (with epicenter at 2cm) also underwent total extended gastrectomy. In terms of preoperative blood levels of tumor markers, CEA positive were N = 19 patients, while Ca19.9 positive were N = 17 patients in the whole EC cohort. The preoperative characteristics of the EC group are listed in Table 1.

In final histopathological examination, N = 2 (2.1%) patients with underlying Barret’s Esophagus had High Grade Dysplasia, whereas invasive carcinomas included Adenocarcinoma (N = 84, 88.4%), Adeno-squamous (N = 2, 2.1%), Squamous Cell Carcinoma (N = 6, 6.3%), and Mixed adeno-neuroendocrine carcinoma (MANEC) (N = 1, 1.1%). Invasive tumors were well-differentiated (G1) in 2/95, moderately differentiated (G2) in 36/95, and poorly differentiated (G3) in 52/95 patients, while in N = 5 differentiation could not be assessed (Gx) or was not applicable (N/A). With respect to tumor infiltration, PNI was positive in 47/95, LVI in 37/95, and PVI in 42/95 patients. Among EAC patients, SRC Status was Positive in N = 20/84, Diffuse type was identified in 15/84 patients, and Intestinal type in 28/84 patients. The histopathological characteristics of the EC group are listed in Table 2.

Overall, in- and out-of-hospital 90-day mortality was 7.4% (N = 7/95). Long-term follow-up was completed in N = 92/95 patients (97%). The follow–up period ranged between 4 and 97 months, with a median 75 months for living cases and 20 months for deceased cases. N = 49/95 patients (51.6%) developed disease recurrence, with N = 46/95 having passed away 4–87 months post-operatively. Thirty-one patients (32.6%) are alive and cancer-free with median survival of 77 months (range 58–97). In total, estimated median OS was 32.5 months (range: 4–97 months), while median DFS was 18.4 months (range: 2–97). The follow-up data of the EC group are listed in Table 3.

We conducted univariate analysis for all the independent histopathological and laboratory prognostic variables of interest. We identified significantly worse OS and DFS associated with PNI (HR: 2.00, *p* = 0.017 and HR: 2.120, *p* = 0.013), LVI (HR: 1.78, *p* = 0.041 and HR: 2.400, *p* = 0.004), PVI (HR: 1.820, *p* = 0.035 and HR: 1.880, *p* = 0.038), and SRC (HR: 1.97, *p* = 0.028 and HR: 2.510, *p* = 0.005) positive status. In terms of the serum CEA and Ca19.9 positive groups, continuous Cox proportional hazard analysis revealed both as significant predictors for recurrence (*p* = 0.000186 and *p* = 0.00405, respectively), but only the Ca19.9 demonstrated significant prognostic significance regarding overall survival (*p* = 0.0159) (Appendix A, Figure 1 and Figure 2).

### 3.2. Allele Frequencies and Genotype Distributions Reflecting the Association between HOTAIR, LINC00951, POLR2E, and HULC Polymorphisms and Cancer Risk Prognostic Factors in EC and EAC Populations

In summary, for all four lncRNAs, the detection of each polymorphism’s distribution was performed in both n = 121 healthy controls and the N = 95 surgically treated EC patients. We subsequently conducted various subset statistical analyses in an effort to identify possible correlations between polymorphisms’ frequency and positive or negative subgroups for each histopathologic and laboratory prognostic risk factors of interest. Histopathologic variables assessed were (a) Histopathologic cell type (EAC versus ESCC), (b) Tumor infiltration in the form of PNI, LVI, and PVI Status in EAC, (c) G Status in EAC, (d) SRC, Diffuse, and Intestinal Type Status in EAC subgroup. We conducted Fisher’s and Chi-squared tests for all the categorical variables. Laboratory variables assessed were preoperative serum tumor markers CEA and Ca19.9, for which we investigated the incidence of the SNPs in either positive or negative for these tumor markers EC subgroups. We also implemented logarithmic transformation, because in our generalised linear model, the continuous variables had non-normal distribution. All SNPs frequencies in patient subgroups were also compared with the frequencies detected within the healthy control group in an effort to compare the underlying molecular basis and identify potential mutual genetic footprint among the healthy controls, cancer positive, and cancer negative subgroups for each prognostic marker.

In detail, for the detection of HOTAIR rs920778, C>T (T/C) was performed in both 121 healthy controls and the 95 surgically treated EC patients’ populations as well as in all the previously defined subgroups (Table 4). In subgroup analysis by histological type (EAC versus ESCC), distribution was not significantly different between the two populations. Similarly, the sub-analysis comparing the healthy controls and EAC patients’ Positive or Negative for PNI, LVI, and PVI yielded no significant correlations. When we explored the SNP’s incidence among various grades of differentiation in EAC patients compared with the incidence in the healthy individuals, we found that both TT and T variants were significantly overrepresented in the Gx–G1 EAC patients compared with the controls (OR: 37.000, *p* = 0.0134 and OR: 6.242, *p* = 0.0155, respectively). When we compared Gx–G1 with G2–G3 EAC subgroups, T allele was also significantly more frequent in the Gx–G1 (OR: 0.2276, *p* = 0.0475) compared with the G2–G3 group. This genetic effect of HOTAIR rs920778 to G-status was also demonstrated when distributions were evaluated with Chi-squared tests (*p* = 0.018, Table 5. T allele was also overrepresented in the SRC positive EAC patients (OR: 2.247, *p* = 0.0278) compared with the healthy controls, while no correlations were detected among Diffuse and Intestinal Type subgroups. When searched for possible association between HOTAIR rs920778 and CEA/Ca19.9 positive EC patient subgroups, we found no correlation with CEA but significantly higher incidence of the CT and T gene variants in Ca19.9 positive groups compared with both the controls (OR: 3.786, *p* = 0.0272 and OR: 2.318, *p* = 0.0317) and Ca19.9 negative group (OR: 4.400, *p* = 0.0194).

Subsequently, we examined LINC00951 rs11752942, A>G (G/A) incidence based on same methodology principles (Table 6). Overall, the SNP’s distribution between either EAC versus ESCC histological subtypes or between EAC patients’ Positive versus Negative subpopulations according to PNI, LVI, and PVI infiltration status did not differ significantly. Likewise, both AG and GG genotypes were found to be equally present in the various grades of EAC tumor differentiation subgroups, as well as in SRC and Diffuse and Intestinal Type Positive EAC patients compared with the controls. While we identified no association with any of the positive EC groups for CEA and Ca19.9 in the Fisher’s tests, when we performed logarithmic transformation for both CEA and Ca19.9, the GG genotype was significantly correlated with Ca19.9 (log-OR: 0.195, *p* = 0.004, Figure 3, Table 7).

We also evaluated the distribution of POLR2E rs3787016, T>C (C/T) polymorphism in both the surgical cancer and healthy control groups (Table 8).

We detected CC and C genotypes significantly more frequently in ESCC compared with the EAC subgroup (OR: 24.600, *p* = 0.0105 and OR: 5.000, *p* = 0.0096, respectively), suggesting that polymorphism may pose a strong risk factor when it comes to esophageal squamous cell carcinogenesis. On the contrary, both CC and C genotypes were found significantly less frequently in EAC patients compared with the healthy controls (OR: 0.2497, *p* = 0.0114 and OR: 0.5778, *p* = 0.0119, respectively), suggesting that it may pose a strong protective risk factor against esophageal adenocarcinoma oncogenesis. We then examined the gene frequencies in EAC subgroups according to PNI, LVI, and PVI infiltration status. CC and C genotypes were significantly underrepresented in subgroups with more dismal prognosis, as in 40/84 PNI Positive, 35/84 LVI Positive, and in 39/84 PVI Positive EAC patients compared with the healthy community subjects, indicating a possible protective role in genetic footprint of EAC invasive potential. This prognostic significance of POLR2E rs3787016 via the LVI pathway was also demonstrated when distributions were evaluated with Chi-squared tests for the whole EC dataset of patients irrespective of the histological subtype (*p* = 0.038, Table 5).

We then performed a subset analysis based on tumor grade of differentiation. This also yielded a significant underrepresentation of the CC (OR: 0.2625, *p* = 0.0119) and C alleles (OR: 0.5828, *p* = 0.0147) in N = 80 EAC-G2/G3 patients compared with the controls. When we investigated SRC Status population, we demonstrated a significantly lower incidence of the CT (OR: 0.2901, *p* = 0.0381) and C (OR: 0.4194, *p* = 0.0342) variants in the SRC positive patients compared with the heathy individuals. C allele was also more infrequent in the Diffuse positive group compared with the controls (OR: 0.3611, *p* = 0.0290), yet not statistically different when compared against the Intestinal positive subgroup. The subset Fisher’s analysis for tumor markers yielded no association in any of the CT/TT, CC/TT, and C/T genetic models with no significant variations among CEA and Ca19.9 Positive/Negative and control subpopulations; however, in the logarithmic transformations, CT/TT was significantly correlated with CEA (log-OR: 2.14, *p* = 0.013, Figure 4, Table 7).

Finally, we sought to explore the prognostic value of HULC rs7763881 by detecting its distribution in both cancer and healthy control groups (Table 9). Overall, the incidence of HULC rs7763881, A>C (C/A) among EAC and ESCC histological subpopulations was not found to be statistically significantly different. Similarly, all subsequent correlations detecting SNP’s variations in subgroups formulated based on PNI, LVI, and PVI infiltration status, G differentiation status, SRC, and Diffuse and Intestinal Type status resulted in no significant associations. Additionally, no molecular association was detected between the HULC rs7763881 and the EC patients with elevated serum CEA or Ca19.9 pre-operatively.

## 4. Discussion

Several risk-assessment models have been designed in an effort to decipher esophageal cancer heterogeneity and contain its unpredictability to ultimately guide a targeted curative treatment and accurately predict outcomes. The Tumor-Node-Metastasis (TNM) classification system is the most widely accepted risk-assessment modality to classify esophageal malignancy and thereby assist in prognostic cancer staging. The most used TNM classification system in the East is the ‘Japanese Classification of Esophageal Cancer’ by the Japanese Esophageal Society (JES) [31], whereas in the West it is the ‘TNM Cancer Staging Manual’ by the AJCC/UICC.

In 2021, Ozawa et al. [32] demonstrated that for all patients included in their study (of note, 93% of whom were ESCCs), the AJCC 8th edition staging system tended to reflect survival more precisely than that of the JES 11th edition, particularly for lower thoracic esophageal tumors. While the main difference between these two prognostic systems is in the definition of the LN stage [33], there are also few notable differences on how these systems utilize prognostic factors such as G and LVI Status and incorporate them in their risk-assessment prognostic models. Despite taking into consideration these parameters, the disease volatility still cannot be exclusively explained or controlled, suggesting that genetic footprint may also contribute to malignant transformation of esophageal epithelium; as such, its role warrants reinvestigation [34].

Emerging literature advocates that lncRNAs and SNPs occurring in their functional region influence the pathophysiology of esophageal oncogenesis [13,35]. Yet, the results obtained so far have been controversial, inconclusive, or limited by small sample-sizes [36]. Moreover, most of the published research investigating this association of lncRNA SNPs in esophageal carcinogenesis has been restricted in Eastern ethnicities where ESCC subtype predominates [37]. Based on current epidemiological evidence, we hypothesized that similar lncRNA SNPs may also interplay in EAC aetiopathogenesis in a western subpopulation such as Greece. To test this hypothesis, we explored the incidence of four lncRNA SNPs on EC surgically treated patients and healthy controls of European/Greek ancestry. We further sought to evaluate the underlying molecular basis for histopathological and laboratory prognostic cancer risk markers by ascertaining the SNPs frequency in these subgroups by conducting subset statistical analysis.

HOX transcript antisense RNA (HOTAIR) is a 2.2-nucleotide lncRNA located in chromosome 12q13.12, transcribed from the homeobox C gene (HOXC) locus [38]. Accumulating research is drawing attention to correlation between HOTAIR’s SNPs and the risk for various cancer types but the results obtained so far have been equivocal [39]. In 2019, in Tian’s review, comprising 107 meta-analyses and 6 genome-wide association studies, HOTAIR rs920778 was rated as strong evidence of true association with ESCC risk for the T allele, yet all included studies were performed on a single ethnic group (Asian) [40]. Conversely, in 2020, Minn et al. [41] concluded that HOTAIR rs920778 did not contribute, either overall or by type cancer incidence, in Japanese population. Taking into account previous evidence, we performed a case-control study analyzing the distribution of HOTAIR rs920778 genotype frequencies in both EC and healthy controls which yielded not significant over-presentation in our EC population in terms of EAC versus ESCC, PNI, LVI, and PVI prognostic variables as opposed to studies implicating HOTAIR with LVI in cancers such as cervical [42,43]. In line with Zhang et al. [21], HOTAIR was significantly associated with G-status in the whole EC cohort, while in EAC T allele was significantly different in Gx–G1 versus controls and G2–G3 versus G1-Gx subgroups. The latter ORs were ambiguous possibly due to the small sample size of the Gx–G1 subgroup and therefore this finding needs to be cautiously taken into consideration. While we found no association with Intestinal or Diffuse types in our cohort, as revealed by Petkevicius et al. in Lithuanian gastric cancer subjects in 2022 [44], T allele was significantly overrepresented in the SRC positive patients when compared with the cancer-free controls. Additionally, while we demonstrated no correlation between HOTAIR SNP and CEA, we identified increased frequency of both CT and T genetic variants in Ca19.9 positive EC patients when compared with both the healthy and the Ca19.9 negative patients, indicating that they may share a common genetic pathway with increased susceptibility for esophagogastric cancer or more dismal prognosis.

LINC00951 is a lncRNA located in chromosome 6p21.2, informally studied as lincRNA-uc003opf.1. A variant genotype of rs11752942 in linc-RNA-uc003opf.1 exon has been reported to be associated with cancer risk. The rs11752942 A>G (G/A) may affect cell proliferation and tumor growth, thereby promoting the susceptibility of ESCC as per Wu et al.’s [45] genotyping results among 52 studied SNPs. LINC00951 rs11752942 was also revealed to be related to head and neck cancers’ incidence in adults [46] as well as in neuroblastoma incidence in children [47] in Asia. Taking these into consideration, we conducted a case-control study to determine possible association between this polymorphism and EC/EAC risk in Greek population. As opposed to studies of Asian background, our analysis investigating the molecular effects of the LINC00951 polymorphism in both the histopathological and laboratory prognostic markers of interest did not uncover significant statistical evidence between the rs11752942 and cancer susceptibility in any of the genetic models AG/AA, GG/AA, and G/A alleles. However, in logarithmic transformation for CEA and Ca19.9, the GG variant was uncovered significantly less frequently in Ca19.9 elevated patients, implying it may affect protectively not only the esophagogastric cancer prognosis but also the molecular behavior of other Ca19.9 producing malignancies [48].

SNP rs3787016 (A>G or its complementary T>C, C/T), localized in the fourth intron of the RNA polymerase II subunit E (POLR2E) lncRNA gene, has been implicated with cancer susceptibility either by predisposing for GC [49], breast and cervical cancer [50], or by protecting against ESCC [51] in Chinese populations. As no study to date has explored the molecular impact of POLR2E rs3787016 in histopathological and laboratory prognostic factors in EC/EAC in the west, we conducted a case-control study in a population of Greek/European ancestry. Our analysis by histological subtype yielded that CC and C allele carriers were significantly higher in ESCC and lower in EAC patients, suggesting that it may pose a risk prognostic factor for the former and protective factor for the latter. These genetic variants were also significantly less frequent in the PNI, LVI, and PVI positive EAC subsets compared with the healthy controls as well as in the LVI positive whole EC cohort. Furthermore, in our subgroup analysis, assessing the molecular basis of the SRC and Diffuse/Intestinal prognostic cancer factors, C allele was also underrepresented in the positive groups. CT variant was found significantly more frequently within the CEA elevated patients, implying it may affect not only the esophagogastric cancer prognosis but also the molecular behavior of other CEA producing malignancies in a similar genetic pattern as KIF26B non-coding RNA in colon cancer [52].

While Kang et al. [51] demonstrated that HULC rs7763881 was a protective prognostic factor against ESCC among male younger patients, Hong et al. [53] suggested an increased GC susceptibility-both studies conducted in Chinese populations. The hepatocellular carcinoma up-regulated lncRNA (HULC) gene is located in chromosome 6p24.3 with two exons and 1638 bp length. Given the literature’s contradictory results, we conducted our case-control study to explore its prognostic significance in EC/EAC genetic footprint in a western ethnicity. Compared with the previous studies, our analysis, investigating the molecular effects of the HULC polymorphism in both the histopathological and laboratory prognostic markers of interest in terms of EAC/ESCC subtypes, PNI, LVI, PVI, G Status, SRC, and Intestinal versus Diffuse subtypes, as well as CEA and Ca19.9, revealed no association in any of the genetic models AC/AA, CC/AA, and C/A alleles. This could be explained by EAC’s predominant subtype prevalence in our cohort or may signify that a true different genetic footprint needs to be confirmed by additional future studies in the west.

Certain limitations apply to this research article. Since it was a hospital-based case-control study with large majority of EC cases and healthy controls from the Attica Region, inherent selection bias may have occurred. The statistical strength of this case-control study may also be limited by the sample size, particularly with respect to the statistical analyses of the subgroups positive or negative to the prognostic factors of interest where the smaller sizes may have impacted the data credibility. We sought to overcome these small-study effects by comparing all the SNPs distributions of the subgroups with our control sample (n = 121) as well as by combining statistical tests to further corroborate our statistically significant results or the trends observed throughout the report. Finally, despite this case-control study being retrospective by definition, all data were extracted from our prospectively collected UGI cancer database following predetermined research protocol to ensure appropriate methodology. Additional study-strengths were the follow-up length with high case-ascertainment enabling us to perform our correlations between SNPs and oncological outcomes such as tumor progression, metastasis, and overall survival.

## 5. Conclusions

In conclusion, HULC rs7763881 was not detected differently in any of the EC prognostic subgroups compared with the healthy community subjects. LINC00951 rs11752942 GG variant was significantly underrepresented in Ca19.9 elevated patient subgroup indicating it may serve as a prognostic marker with protective potential not only for esophagogastric cancer but also for other Ca19.9 secretory malignancies. HOTAIR rs920778 TT and T genotypes were significantly associated with prognostic factors as G differentiation grade and SRC status, whereas CT and T genotypes with Ca19.9 elevated patient subgroup suggesting it may serve as a potential therapeutic suppression target against esophagogastric cancer in addition to estimate prognosis in Ca19.9 secretory malignancies. Regarding POLR2E rs3787016, CC and C genotypes were significantly correlated with histological subtypes such as ESCC, EAC, SRC and Diffuse, as well as with prognostic variables in the form of PNI, LVI, and PVI, whereas CT variant was associated with CEA. This indicates that it may be able to evaluate esophagogastric cancer predisposition and predict response to treatment and prognosis in CEA secretory malignancies in the future.

Overall, the present study demonstrates that lncRNAs’ LINC00951, HOTAIR and POLR2E polymorphisms may genetically influence and as such, may explain a fraction of EC and EAC molecular basis. Implementation of these genetic models as part of the clinical and pathological risk-assessment process may add to the efficiency and efficacy of the current utilized prognostic models. Prospective multicenter studies with larger sample-size are required to validate these findings.

## Figures and Tables

**Figure 1 cancers-16-00537-f001:**
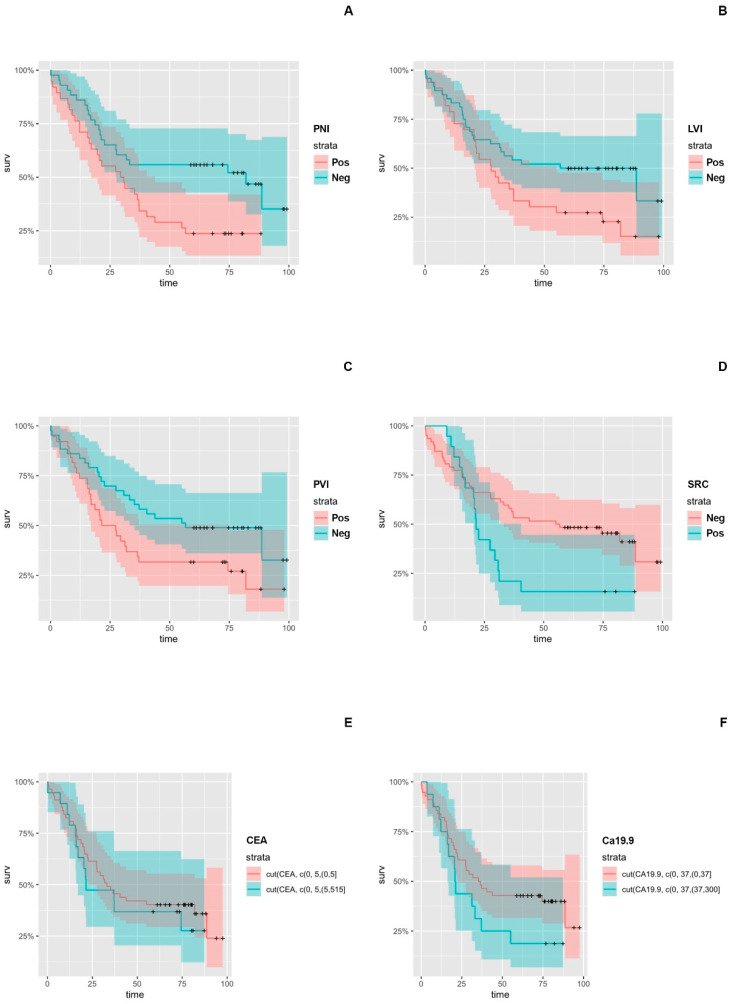
Kaplan–Meier Curves for histopathological and laboratory variables with statistical prognostic significance regarding Overall Survival (OS) in Esophageal Cancer (EC) and Esophageal adenocarcinoma (EAC) Populations: (**A**) PNI, (**B**) LVI, (**C**) PVI, (**D**) SRC, (**E**) CEA, (**F**) Ca19.9.

**Figure 2 cancers-16-00537-f002:**
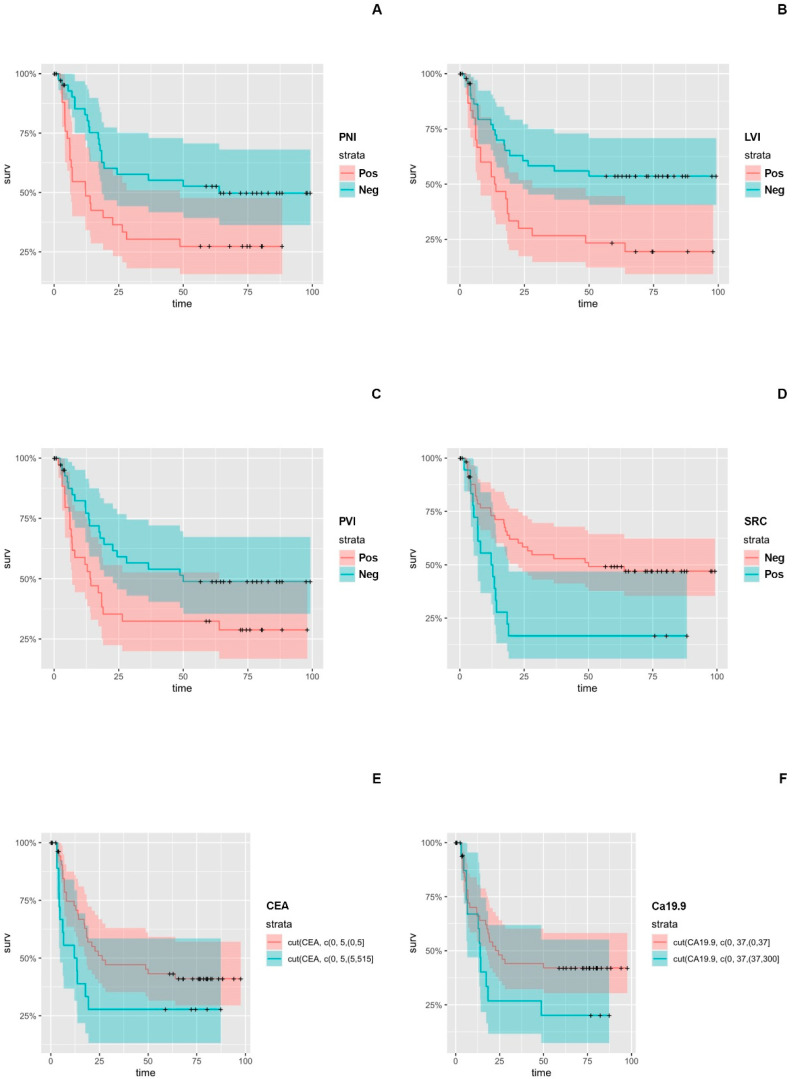
Kaplan-Meier Curves for histopathological and laboratory variables with statistical prognostic significance regarding Disease-Free Survival (DFS) in Esophageal Cancer (EC) and Esophageal adenocarcinoma (EAC) Populations: (**A**) PNI, (**B**) LVI, (**C**) PVI, (**D**) SRC, (**E**) CEA, (**F**) Ca19.9.

**Figure 3 cancers-16-00537-f003:**
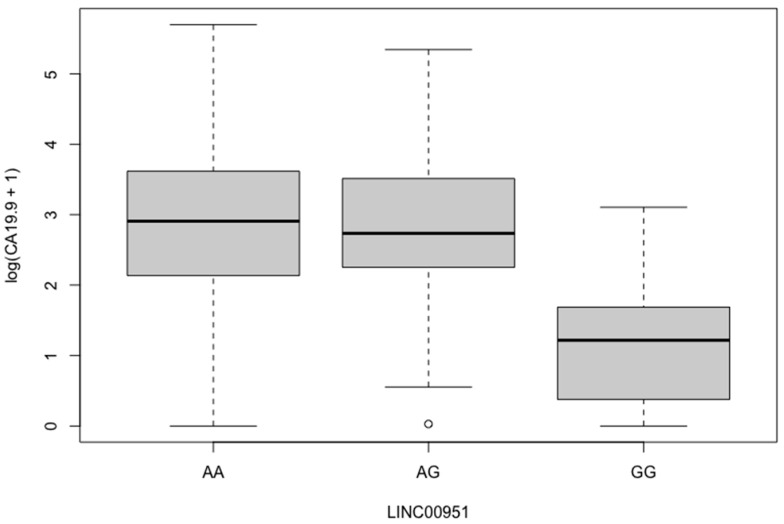
Logarithmic transformation demonstrating that LINC00951 rs11752942 GG/AA genotype is significantly correlated with Ca19.9 (log-OR: 0.195, *p* = 0.004).

**Figure 4 cancers-16-00537-f004:**
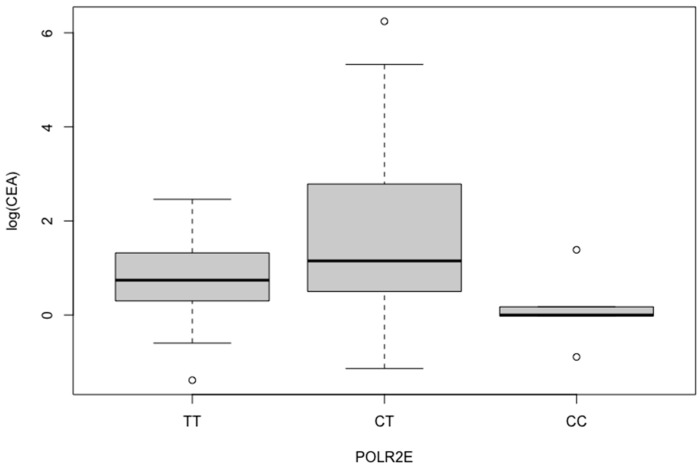
Logarithmic transformation demonstrating that POLR2E rs3787016 CT/TT is significantly correlated with CEA (log-OR: 2.14, *p* = 0.013).

**Table 1 cancers-16-00537-t001:** Demographics-Preoperative Data for the Esophageal Cancer (EC) cohort (N = 95 patients).

Variables	Value N (%)
Age (Mean ± SD, years)/Median (range), years	62.9 ± 11.39/63 (27–83)
Gender: Male/Female	Ν = 86 (90.5%)/Ν = 9 (9.5%)
ASA Score:	
I	Ν = 31 (32.6%)
II	Ν = 46 (48.4%)
III	Ν = 15 (15.8%)
IV	Ν = 3 (3.2%)
Preoperative Tumor Markers:	
CEA positive/negative/NR	N = 19 (20%)/59 (62%)/17 (18%)
Ca19.9 positive/negative/NR	N = 17 (18%)/59 (62%)/19 (20%)
Chemotherapy/Chemoradiotherapy:	
Neoadjuvant	N = 24 (25.3%)
Adjuvant	N = 55 (57.9%)
Tumor Location:	
MT Esophagus	N = 2 (2.1%)
LT Esophagus	N = 8 (8.4%)
EGJ-Siewert I	N = 20 (21.1%)
EGJ-Siewert II	N = 44 (46.3%)
EGJ-Siewert III	N = 21 (22.1%)
Operative technique:	
IL 2f-2s-esophagectomy	N = 48 (50.5%)
MK 3f-3s-esophagectomy	N = 19 (20%)
Total extended gastrectomy	N = 28 (29.5%)

Notes: Carcinoembryonic Antigen (CEA) defined as Positive > 5 ng/mL and Negative < 5 ng/mL, Ca19.9 defined as Positive > 37 U/mL and Negative < 37 U/mL. MT: Middle Thoracic Esophagus, LT: Lower Thoracic Esophagus, EsophagoGastric Junction (EGJ), 2f: 2-field standard lymph-node dissection, 2s: two stage (laparotomic/thoracotomic), 3f: 3-field standard lymph-node dissection, 3s: three stage (laparotomic/thoracotomic/left neck), IL: Ivor Lewis, MK: McKeown. NR: Not reported.

**Table 2 cancers-16-00537-t002:** Final Histopathological Data for the Esophageal Cancer (EC) cohort (N = 95 patients).

Variables	Value N (%)
Histological Type:	
Adenocarcinoma (EAC)	N = 84 (88.4%)
Adeno-squamous	N = 2 (2.1%)
Squamous Cell Carcinoma	N = 6 (6.3%)
MANEC	N = 1 (1.1%)
High Grade Dysplasia	N = 2 (2.1%)
Tumor Differentiation:	
Well-differentiated (G1)	N = 2 (2.1%)
Moderately differentiated (G2)	N = 36 (37.9%)
Poorly differentiated (G3)	N = 52 (54.7%)
Cannot be assessed (Gx) or N/A	N = 5 (5.3%)
Type of Tumor Infiltration:	
Perineural Invasion (PNI) positive	N = 45 (47.4%)
Lymphovascular Invasion (LVI) positive	N = 37 (38.9%)
Perivascular Invasion (PVI) positive	N = 42 (44.2%)
Signet Ring Cell (SRC) Status in N = 84 EAC Patients:	
Positive/Negative	N = 20 (24%)/64 (76%)
Diffuse/ Intestinal Type in N = 84 EAC Patients:	N = 15 (18%)/28 (33.3%)
Final pathological TNM staging:	
0	N = 4 (4.2%)
I	Ν = 10 (10.5%)
II	Ν = 18 (19%)
III	Ν = 42 (44.2%)
IV	N = 21 (22.1%)
Tumor (T) status:	
pT0	N = 1 (1.1%)
pTis	Ν = 3 (3.2%)
pT1	N = 7 (7.4%)
pT2	N = 19 (20%)
pT3	N = 54 (56.8%)
pT4	N = 11 (11.5%)
Lymph Node (N) status:	
N0	Ν = 29 (30.5%)
N1	Ν = 13 (13.7%)
N2	Ν = 22 (23.2%)
N3	Ν = 31 (32.6%)
Lymph node harvest:	
>15	N = 87 (91.6%)
<15	N = 8 (8.4%)
Resection Status:	
R0	Ν = 86 (90.5%)
R1	Ν = 9 (9.5%)
R2	Ν = 0 (0%)
Circumferential Resection Margin (CRM):	
Negative	Ν = 85 (89.5%)
Positive	Ν = 10 (10.5%)

Notes: Adenocarcinoma (EAC), MANEC: Mixed adeno-neuroendocrine carcinoma, N/A: Not Applicable.

**Table 3 cancers-16-00537-t003:** Postoperative Follow-up Outcomes for the Esophageal Cancer (EC) cohort (N = 95 patients).

Variables	Value N (%)
Clavien-Dindo Complications (90-day):	
None	Ν = 48 (50.5%)
I	N = 4 (4.2%)
II	N = 19 (20%)
IIIa	N = 14 (14.7%)
IIIb	N = 1 (1.1%)
IVa	N = 2 (2.1%)
IVb	N = 0 (0%)
V	N = 7 (7.4%)
Type of 1st disease progression:	
Local recurrence	N = 2 (2.1%)
Regional LN metastasis	N = 10 (10.5%)
Distant Metastasis	N = 32 (33.7%)
Combined	N = 5 (5.3%)
Median Disease-Free Survival (months, range)	18.4 (2–97)
Median Overall Survival (months, range)	32.5 (4–97)
Median Length of Follow-up (months, range)	36 (4–97)

**Table 4 cancers-16-00537-t004:** Allele frequencies and genotype distributions demonstrating the association between HOTAIR polymorphism and cancer risk prognostic factors in Esophageal Cancer (EC) and Esophageal adenocarcinoma (EAC) populations—Bold values denote statistically significant associations.

Genotype: HOTAIR-SNP: rs920778, C>T (T/C)				
EC Population (N = 95)	Adenocarcinoma, N = 84 (%)	Squamous Cell Carcinoma, N = 6 (%)	OR (95% CI)	*p* Value
CC	43 (51.2)	4 (66.7)	1.00 (Ref.)	1.00 (Ref.)
CT	33 (39.3)	2 (33.3)	0.6515 (0.1124–3.777)	1.0000
TT	8 (9.5)	0 (0)	0.5686 (0.02793–11.577)	1.0000
C allele	119 (70.9)	10 (83.4)	1.00 (Ref.)	1.00 (Ref.)
T allele	49 (29.1)	2 (16.6)	0.4857 (0.1026–2.299)	0.5135
EAC subpopulation (N = 84/95)-Perineural Invasion (PNI) Status	Control Group, n = 121 (%)	Positive, N = 40 (%)	OR (95% CI)	*p* value
CC	74 (61.2)	21 (52.5)	1.00 (Ref.)	1.00 (Ref.)
CT	43 (35.5)	16 (40)	1.311 (0.6185–2.780)	0.5614
TT	4 (3.3)	3 (7.5)	2.643 (0.5477–12.752)	0.3507
C allele	191 (79)	58 (72.5)	1.00 (Ref.)	1.00 (Ref.)
T allele	51 (21)	22 (27.5)	1.421 (0.7953–2.537)	0.2807
EAC subpopulation (N = 84/95)-Perineural Invasion (PNI) Status	Negative, N = 44 (%)	Positive, N = 40 (%)	OR (95% CI)	*p* value
CC	22 (50)	21 (52.5)	1.00 (Ref.)	1.00 (Ref.)
CT	17 (38.6)	16 (40)	0.9860 (0.3978–2.444)	1.0000
TT	5 (11.4)	3 (7.5)	0.6286 (0.1332–2.966)	0.7071
C allele	61 (69.3)	58 (72.5)	1.00 (Ref.)	1.00 (Ref.)
T allele	27 (30.7)	22 (27.5)	0.8570 (0.4394–1.671)	0.7346
EAC subpopulation (N = 84/95)-Lymphovascular Invasion (LVI) Status	Control Group, n = 121 (%)	Positive, N = 35 (%)	OR (95% CI)	*p* value
CC	74 (61.2)	17 (48.6)	1.00 (Ref.)	1.00 (Ref.)
CT	43 (35.5)	16 (45.7)	1.620 (0.7429–3.532)	0.2332
TT	4 (3.3)	2 (5.7)	2.176 (0.3678–12.878)	0.3342
C allele	191 (79)	50 (71.5)	1.00 (Ref.)	1.00 (Ref.)
T allele	51 (21)	20 (28.5)	1.498 (0.8191–2.740)	0.1977
EAC subpopulation (N = 84/95)-Lymphovascular Invasion (LVI) Status	Negative, N = 49 (%)	Positive, N = 35 (%)	OR (95% CI)	*p* value
CC	26 (53.1)	17 (48.6)	1.00 (Ref.)	1.00 (Ref.)
CT	17 (34.7)	16 (45.7)	1.439 (0.5756–3.600)	0.4889
TT	6 (12.2)	2 (5.7)	0.5098 (0.09188–2.829)	0.6936
C allele	69 (70.5)	50 (71.5)	1.00 (Ref.)	1.00 (Ref.)
T allele	29 (29.5)	20 (28.5)	0.9517 (0.4840–1.871)	1.0000
EAC subpopulation (N = 84/95)-Perivascular Invasion (PVI) Status	Control Group, n = 121 (%)	Positive, N = 39 (%)	OR (95% CI)	*p* value
CC	74 (61.2)	22 (56.4)	1.00 (Ref.)	1.00 (Ref.)
CT	43 (35.5)	15 (38.5)	1.173 (0.5507–2.500)	0.7004
TT	4 (3.3)	2 (5.1)	1.682 (0.2884–9.808)	0.6237
C allele	191 (79)	59 (75.7)	1.00 (Ref.)	1.00 (Ref.)
T allele	51 (21)	19 (24.3)	1.206 (0.6603–2.203)	0.5326
EAC subpopulation (N = 84/95)-Perivascular Invasion (PVI) Status	Negative, N = 45 (%)	Positive, N = 39 (%)	OR (95% CI)	*p* value
CC	21 (46.7)	22 (56.4)	1.00 (Ref.)	1.00 (Ref.)
CT	18 (40)	15 (38.5)	0.7955 (0.3203–1.975)	0.6503
TT	6 (13.3)	2 (5.1)	0.3182 (0.05762–1.757)	0.2553
C allele	60 (66.7)	59 (75.7)	1.00 (Ref.)	1.00 (Ref.)
T allele	30 (33.3)	19 (24.3)	0.6441 (0.3270–1.269)	0.2352
EAC subpopulation (N = 84/95)-Differentiation Grade (G) Status	Control Group, n = 121 (%)	G2–G3, N = 80 (%)	OR (95% CI)	*p* value
CC	74 (61.2)	42 (52.5)	1.00 (Ref.)	1.00 (Ref.)
CT	43 (35.5)	32 (40)	1.311 (0.7239–2.375)	0.4472
TT	4 (3.3)	6 (7.5)	2.643 (0.7053–9.902)	0.1785
C allele	191 (79)	116 (72.5)	1.00 (Ref.)	1.00 (Ref.)
T allele	51 (21)	44 (27.5)	1.421 (0.8926–2.261)	0.1509
EAC subpopulation (N = 84/95)-Differentiation Grade (G) Status	Control Group, n = 121 (%)	Gx–G1, N = 4 (%)	OR (95% CI)	*p* value
CC	74 (61.2)	1 (25)	1.00 (Ref.)	1.00 (Ref.)
CT	43 (35.5)	1 (25)	1.721 (0.1049–28.240)	1.0000
TT	4 (3.3)	2 (50)	**37.000 (2.739–499.87)**	**0.0134**
C allele	191 (79)	3 (37.5)	1.00 (Ref.)	1.00 (Ref.)
T allele	51 (21)	5 (62.5)	**6.242 (1.443–27.003)**	**0.0155**
EAC subpopulation (N = 84/95)-Differentiation Grade (G) Status	Gx–G1, N = 4 (%)	G2–G3, N = 80 (%)	OR (95% CI)	*p* value
CC	1 (25)	42 (52.5)	1.00 (Ref.)	1.00 (Ref.)
CT	1 (25)	32 (40)	0.7619 (0.04585–12.660)	1.0000
TT	2 (50)	6 (7.5)	0.07143 (0.005583–0.9138)	0.0605
C allele	3 (37.5)	116 (72.5)	1.00 (Ref.)	1.00 (Ref.)
T allele	5 (62.5)	44 (27.5)	**0.2276 (0.05216–0.9930)**	**0.0475**
EAC subpopulation (N = 84/95)-Differentiation Grade (G) Status	G2, N = 31 (%)	G3, N = 49 (%)	OR (95% CI)	*p* value
CC	16 (51.6)	26 (53.1)	1.00 (Ref.)	1.00 (Ref.)
CT	14 (45.2)	18 (36.7)	0.7912 (0.3103–2.017)	0.6412
TT	1 (3.2)	5 (10.2)	3.077 (0.3289–28.790)	0.4022
C allele	46 (74.2)	70 (71.4)	1.00 (Ref.)	1.00 (Ref.)
T allele	16 (25.8)	28 (28.6)	1.150 (0.5607–2.359)	0.7209
EAC subpopulation (N = 84/95)-Signet Ring Cell (SRC) Status	Control Group, n = 121 (%)	SRC positive, N = 20 (%)	OR (95% CI)	*p* value
CC	74 (61.2)	7 (35)	1.00 (Ref.)	1.00 (Ref.)
CT	43 (35.5)	11 (55)	2.704 (0.9755–7.497)	0.0696
TT	4 (3.3)	2 (10)	5.286 (0.8176–34.173)	0.1151
C allele	191 (79)	25 (62.5)	1.00 (Ref.)	1.00 (Ref.)
T allele	51 (21)	15 (37.5)	**2.247 (1.104–4.575)**	**0.0278**
EAC subpopulation (N = 84/95)-Lauren Classification Status	Control Group, n = 121 (%)	Diffuse Type positive, N = 15 (%)	OR (95% CI)	*p* value
CC	74 (61.2)	8 (53.4)	1.00 (Ref.)	1.00 (Ref.)
CT	43 (35.5)	5 (33.3)	1.076 (0.3308–3.497)	1.0000
TT	4 (3.3)	2 (13.3)	4.625 (0.7287–29.354)	0.1365
C allele	191 (79)	21 (70.05)	1.00 (Ref.)	1.00 (Ref.)
T allele	51 (21)	9 (29.95)	1.605 (0.6929–3.718)	0.2533
EAC subpopulation (N = 84/95)-Lauren Classification Status	Intestinal Type positive, N = 28 (%)	Diffuse Type positive, N = 15 (%)	OR (95% CI)	*p* value
CC	17 (60.8)	8 (53.4)	1.00 (Ref.)	1.00 (Ref.)
CT	9 (32.1)	5 (33.3)	1.181 (0.2972–4.689)	1.0000
TT	2 (7.1)	2 (13.3)	2.125 (0.2518–17.936)	0.5920
C allele	43 (76.85)	21 (70.05)	1.00 (Ref.)	1.00 (Ref.)
T allele	13 (23.15)	9 (29.95)	1.418 (0.5228–3.844)	0.6052
EC-Preop Tumor Marker CEA	Control Group, n = 121 (%)	CEA positive, (>5 ng/mL), N = 19 (%)	OR (95% CI)	*p* value
CC	74 (61.2)	11 (57.9)	1.00 (Ref.)	1.00 (Ref.)
CT	43 (35.5)	7 (36.8)	1.095 (0.3950–3.036)	1.0000
TT	4 (3.3)	1 (5.3)	1.682 (0.1718–16.468)	0.5196
C allele	191 (79)	29 (76.3)	1.00 (Ref.)	1.00 (Ref.)
T allele	51 (21)	9 (23.7)	1.162 (0.5174–2.611)	0.6759
EC-Preop Tumor Marker CEA	CEA negative (<5 ng/mL), N = 59 (%)	CEA positive, (>5 ng/mL), N = 19 (%)	OR (95% CI)	*p* value
CC	30 (50.8)	11 (57.9)	1.00 (Ref.)	1.00 (Ref.)
CT	24 (40.7)	7 (36.8)	0.7955 (0.2676–2.364)	0.7865
TT	5 (8.5)	1 (5.3)	0.5455 (0.05715–5.206)	1.0000
C allele	84 (71.2)	29 (76.3)	1.00 (Ref.)	1.00 (Ref.)
T allele	34 (28.8)	9 (23.7)	0.7667 (0.3285–1.790)	0.6771
EC-Preop Tumor Marker Ca19.9	Control Group, n = 121 (%)	Ca19.9 positive, (>37 U/mL), N = 17 (%)	OR (95% CI)	*p* value
CC	74 (61.2)	5 (29.4)	1.00 (Ref.)	1.00 (Ref.)
CT	43 (35.5)	11 (64.7)	**3.786 (1.233–11.630)**	**0.0272**
TT	4 (3.3)	1 (5.9)	3.700 (0.3453–39.646)	0.3162
C allele	191 (79)	21 (61.8)	1.00 (Ref.)	1.00 (Ref.)
T allele	51 (21)	13 (38.2)	**2.318 (1.087–4.946)**	**0.0317**
EC-Preop Tumor Marker Ca19.9	Ca19.9 negative (<37 U/mL), N = 59 (%)	Ca19.9 positive, (>37 U/mL), N = 17 (%)	OR (95% CI)	*p* value
CC	36 (61)	5 (29.4)	1.00 (Ref.)	1.00 (Ref.)
CT	18 (30.5)	11 (64.7)	**4.400 (1.326–14.598)**	**0.0194**
TT	5 (8.5)	1 (5.9)	1.440 (0.1384–14.987)	1.0000
C allele	90 (76.3)	21 (61.8)	1.00 (Ref.)	1.00 (Ref.)
T allele	28 (23.7)	13 (38.2)	1.990 (0.8838–4.480)	0.1239

**Table 5 cancers-16-00537-t005:** Chi-Squared tests demonstrating the association between the four polymorphisms and the histopathological cancer risk prognostic factors of interest in Esophageal Cancer (EC) and Esophageal adenocarcinoma (EAC) populations—Bold values denote statistically significant associations.

**Chi-Squared Tests: EAC Subpopulation (N = 84/95)—*p* Values**				
	HOTAIR	LINC00951	POLR2E	HULC
PNI	0.834	0.603	0.205	0.811
LVI	0.444	0.446	0.105	0.798
PVI	0.391	0.442	0.953	0.172
G Status	**0.018**	0.800	0.873	0.658
SRC Status	0.345	0.568	0.142	0.748
Intestinal/Diffuse	0.841	0.739	0.318	0.939
**Chi-Squared tests: EC complete dataset (N = 95)—*p* values**				
	HOTAIR	LINC00951	POLR2E	HULC
EAC/ESCC	0.644	0.494	**0.001**	0.557
PNI	0.786	0.571	0.195	0.894
LVI	0.619	0.350	**0.038**	0.520
PVI	0.354	0.676	0.723	0.307
G Status	0.105	0.673	0.672	0.735
SRC Status	0.197	0.488	0.133	0.628
Intestinal/Diffuse	0.841	0.739	0.318	0.939

**Table 6 cancers-16-00537-t006:** Allele frequencies and genotype distributions demonstrating the association between LINC00951 polymorphism and cancer risk prognostic factors in Esophageal Cancer (EC) and Esophageal adenocarcinoma (EAC) populations.

Genotype: LINC00951 SNP: rs11752942, A>G (G/A)				
EC Population (N = 95)	Adenocarcinoma (EAC), N = 84 (%)	Squamous Cell Carcinoma, N = 6 (%)	OR (95% CI)	*p* Value
AA	39 (46.4)	2 (33.3)	1.00 (Ref.)	1.00 (Ref.)
AG	37 (44.1)	4 (66.7)	2.108 (0.3640–12.209)	0.6755
GG	8 (9.5)	0 (0)	0.9294 (0.04078–21.180)	1.0000
A allele	115 (68.5)	8 (66.7)	1.00 (Ref.)	1.00 (Ref.)
G allele	53 (31.5)	4 (33.3)	1.085 (0.3127–3.764)	1.0000
EAC subpopulation (N = 84/95)-Perineural Invasion (PNI) Status	Control Group, n = 121 (%)	Positive, N = 40 (%)	OR (95% CI)	*p* value
AA	47 (38.9)	19 (47.5)	1.00 (Ref.)	1.00 (Ref.)
AG	58 (47.9)	16 (40)	0.6824 (0.3164–1.472)	0.3372
GG	16 (13.2)	5 (12.5)	0.7730 (0.2480–2.410)	0.7831
A allele	152 (62.9)	54 (67.5)	1.00 (Ref.)	1.00 (Ref.)
G allele	90 (37.1)	26 (32.5)	0.8132 (0.4759–1.389)	0.5028
EAC subpopulation (N = 84/95)-Perineural Invasion (PNI) Status	Negative, N = 44 (%)	Positive, N = 40 (%)	OR (95% CI)	*p* value
AA	20 (45.5)	19 (47.5)	1.00 (Ref.)	1.00 (Ref.)
AG	21 (47.7)	16 (40)	0.8020 (0.3247–1.981)	0.6529
GG	3 (6.8)	5 (12.5)	1.754 (0.3674–8.377)	0.7008
A allele	61 (69.4)	54 (67.5)	1.00 (Ref.)	1.00 (Ref.)
G allele	27 (30.6)	26 (32.5)	1.088 (0.5671–2.087)	0.8685
EAC subpopulation (N = 84/95)-Lymphovascular Invasion (LVI) Status	Control Group, n = 121 (%)	Positive, N = 35 (%)	OR (95% CI)	*p* value
AA	47 (38.9)	15 (42.9)	1.00 (Ref.)	1.00 (Ref.)
AG	58 (47.9)	15 (42.9)	0.8103 (0.3595–1.827)	0.6800
GG	16 (13.2)	5 (14.2)	0.9792 (0.3068–3.125)	1.0000
A allele	152 (62.9)	45 (64.4)	1.00 (Ref.)	1.00 (Ref.)
G allele	90 (37.1)	25 (35.6)	0.9383 (0.5391–1.633)	0.8885
EAC subpopulation (N = 84/95)-Lymphovascular Invasion (LVI) Status	Negative, N = 49 (%)	Positive, N = 35 (%)	OR (95% CI)	*p* value
AA	24 (49)	15 (42.9)	1.00 (Ref.)	1.00 (Ref.)
AG	22 (44.9)	15 (42.9)	1.091 (0.4345–2.739)	1.0000
GG	3 (6.1)	5 (14.2)	2.667 (0.5546–12.823)	0.2581
A allele	70 (71.5)	45 (64.4)	1.00 (Ref.)	1.00 (Ref.)
G allele	28 (28.5)	25 (35.6)	1.389 (0.7202–2.678)	0.4001
EAC subpopulation (N = 84/95)-Perivascular Invasion (PVI) Status	Control Group, n = 121 (%)	Positive, N = 39 (%)	OR (95% CI)	*p* value
AA	47 (38.9)	21 (53.8)	1.00 (Ref.)	1.00 (Ref.)
AG	58 (47.9)	15 (38.5)	0.5788 (0.2690–1.246)	0.1798
GG	16 (13.2)	3 (7.7)	0.4196 (0.1103–1.597)	0.2529
A allele	152 (62.9)	57 (73.1)	1.00 (Ref.)	1.00 (Ref.)
G allele	90 (37.1)	21 (26.9)	0.6222 (0.3539–1.094)	0.1029
EAC subpopulation (N = 84/95)-Perivascular Invasion (PVI) Status	Negative, N = 45 (%)	Positive, N = 39 (%)	OR (95% CI)	*p* value
AA	18 (40)	21 (53.8)	1.00 (Ref.)	1.00 (Ref.)
AG	22 (48.9)	15 (38.5)	0.5844 (0.2353–1.451)	0.2616
GG	5 (11.1)	3 (7.7)	0.5143 (0.1076–2.457)	0.4614
A allele	58 (64.5)	57 (73.1)	1.00 (Ref.)	1.00 (Ref.)
G allele	32 (35.5)	21 (26.9)	0.6678 (0.3448–1.293)	0.2480
EAC subpopulation (N = 84/95)-Differentiation Grade (G) Status	Control Group, n = 121 (%)	G2–G3, N = 80 (%)	OR (95% CI)	*p* value
AA	47 (38.9)	37 (46.2)	1.00 (Ref.)	1.00 (Ref.)
AG	58 (47.9)	35 (43.8)	0.7665 (0.4201–1.399)	0.4443
GG	16 (13.2)	8 (10)	0.6351 (0.2451–1.646)	0.4819
A allele	152 (62.9)	109 (68.1)	1.00 (Ref.)	1.00 (Ref.)
G allele	90 (37.1)	51 (31.9)	0.7902 (0.5178–1.206)	0.2874
EAC subpopulation (N = 84/95)-Differentiation Grade (G) Status	Control Group, n = 121 (%)	Gx–G1, N = 4 (%)	OR (95% CI)	*p* value
AA	47 (38.9)	2 (50)	1.00 (Ref.)	1.00 (Ref.)
AG	58 (47.9)	2 (50)	0.8103 (0.1099–5.975)	1.0000
GG	16 (13.2)	0 (0)	0.5758 (0.02624–12.633)	1.0000
A allele	152 (62.9)	6 (75)	1.00 (Ref.)	1.00 (Ref.)
G allele	90 (37.1)	2 (25)	0.5630 (0.1112–2.850)	0.7140
EAC subpopulation (N = 84/95)-Differentiation Grade (G) Status	Gx–G1, N = 4 (%)	G2–G3, N = 80 (%)	OR (95% CI)	*p* value
AA	2 (50)	37 (46.2)	1.00 (Ref.)	1.00 (Ref.)
AG	2 (50)	35 (43.8)	0.9459 (0.1262–7.090)	1.0000
GG	0 (0)	8 (10)	1.133 (0.04969–25.849)	1.0000
A allele	6 (75)	109 (68.1)	1.00 (Ref.)	1.00 (Ref.)
G allele	2 (25)	51 (31.9)	1.404 (0.2737–7.199)	1.0000
EAC subpopulation (N = 84/95)-Differentiation Grade (G) Status	G2, N = 31 (%)	G3, N = 49 (%)	OR (95% CI)	*p* value
AA	13 (41.95)	24 (49)	1.00 (Ref.)	1.00 (Ref.)
AG	13 (41.95)	22 (44.9)	0.9167 (0.3502–2.400)	1.0000
GG	5 (16.1)	3 (6.1)	0.3250 (0.06675–1.582)	0.2351
A allele	39 (62.9)	70 (71.5)	1.00 (Ref.)	1.00 (Ref.)
G allele	23 (37.1)	28 (28.5)	0.6783 (0.3448–1.334)	0.2977
EAC subpopulation (N = 84/95)-Signet Ring Cell (SRC) Status	Control Group, n = 121 (%)	SRC positive, N = 20 (%)	OR (95% CI)	*p* value
AA	47 (38.9)	9 (45)	1.00 (Ref.)	1.00 (Ref.)
AG	58 (47.9)	8 (40)	0.7203 (0.2578–2.012)	0.6046
GG	16 (13.2)	3 (15)	0.9792 (0.2355–4.071)	1.0000
A allele	152 (62.9)	26 (65)	1.00 (Ref.)	1.00 (Ref.)
G allele	90 (37.1)	14 (35)	0.9094 (0.4515–1.832)	0.8608
EAC subpopulation (N = 84/95)-Lauren Classification Status	Control Group, n = 121 (%)	Diffuse Type positive, N = 15 (%)	OR (95% CI)	*p* value
AA	47 (38.9)	7 (46.7)	1.00 (Ref.)	1.00 (Ref.)
AG	58 (47.9)	8 (53.3)	0.9261 (0.3129–2.741)	1.0000
GG	16 (13.2)	0 (0)	0.1919 (0.01037–3.550)	0.3387
A allele	152 (62.9)	22 (73.35)	1.00 (Ref.)	1.00 (Ref.)
G allele	90 (37.1)	8 (26.65)	0.6141 (0.2624–1.437)	0.3159
EAC subpopulation (N = 84/95)-Lauren Classification Status	Intestinal Type positive, N = 28 (%)	Diffuse Type positive, N = 15 (%)	OR (95% CI)	*p* value
AA	12 (42.9)	7 (46.7)	1.00 (Ref.)	1.00 (Ref.)
AG	13 (46.4)	8 (53.3)	1.055 (0.2925–3.805)	1.0000
GG	3 (10.7)	0 (0)	0.2381 (0.01074–5.281)	0.5227
A allele	37 (66.1)	22 (73.35)	1.00 (Ref.)	1.00 (Ref.)
G allele	19 (33.9)	8 (26.65)	0.7081 (0.2657–1.888)	0.6271
EC-Preop Tumor Marker CEA	Control Group, n = 121 (%)	CEA positive, (>5 ng/mL), N = 19 (%)	OR (95% CI)	*p* value
AA	47 (38.9)	11 (57.9)	1.00 (Ref.)	1.00 (Ref.)
AG	58 (47.9)	7 (36.8)	0.5157 (0.1854–1.434)	0.2139
GG	16 (13.2)	1 (5.3)	0.2670 (0.03190–2.235)	0.2765
A allele	152 (62.9)	29 (76.3)	1.00 (Ref.)	1.00 (Ref.)
G allele	90 (37.1)	9 (23.7)	0.5241 (0.2374–1.157)	0.1435
EC-Preop Tumor Marker CEA	CEA negative (<5 ng/mL), N = 59 (%)	CEA positive, (>5 ng/mL), N = 19 (%)	OR (95% CI)	*p* value
AA	24 (40.7)	11 (57.9)	1.00 (Ref.)	1.00 (Ref.)
AG	30 (50.8)	7 (36.8)	0.5091 (0.1713–1.513)	0.2803
GG	5 (8.5)	1 (5.3)	0.4364 (0.04539–4.195)	0.6514
A allele	78 (66.1)	29 (76.3)	1.00 (Ref.)	1.00 (Ref.)
G allele	40 (33.9)	9 (23.7)	0.6052 (0.2614–1.401)	0.3155
EC-Preop Tumor Marker Ca19.9	Control Group, n = 121 (%)	Ca19.9 positive, (>37 U/mL), N = 17 (%)	OR (95% CI)	*p* value
AA	47 (38.9)	9 (52.9)	1.00 (Ref.)	1.00 (Ref.)
AG	58 (47.9)	8 (47.1)	0.7203 (0.2578–2.012)	0.6046
GG	16 (13.2)	0 (0)	0.1515 (0.008344–2.751)	0.1924
A allele	152 (62.9)	26 (76.5)	1.00 (Ref.)	1.00 (Ref.)
G allele	90 (37.1)	8 (23.5)	0.5197 (0.2256–1.197)	0.1302
EC-Preop Tumor Marker Ca19.9	Ca19.9 negative (<37 U/mL), N = 59 (%)	Ca19.9 positive, (>37 U/mL), N = 17 (%)	OR (95% CI)	*p* value
AA	26 (44.1)	9 (52.9)	1.00 (Ref.)	1.00 (Ref.)
AG	27 (45.8)	8 (47.1)	0.8560 (0.2865–2.557)	1.0000
GG	6 (10.1)	0 (0)	0.2146 (0.01100–4.187)	0.3090
A allele	79 (67)	26 (76.5)	1.00 (Ref.)	1.00 (Ref.)
G allele	39 (33)	8 (23.5)	0.6233 (0.2584–1.504)	0.3998

**Table 7 cancers-16-00537-t007:** Logarithmic Transformation tests demonstrating the association between the four polymorphisms and the laboratory cancer risk prognostic factors of interest in Esophageal Cancer (EC) Population—Bold values denote statistically significant associations.

**CEA**				
	log-odds ratio	LCI	HCI	*p* value
HOTAIRCT	0.924	0.491	1.74	0.807
HOTAIRTT	1.01	0.318	3.24	0.981
LINC00951AG	0.671	0.362	1.24	0.209
LINC00951GG	0.508	0.16	1.61	0.254
POLR2ECT	2.14	1.19	3.84	**0.013**
POLR2ECC	0.521	0.158	1.72	0.288
HULCAC	1.1	0.541	2.25	0.788
HULCCC	1.1	0.498	2.43	0.814
**Ca19.9**				
	log-odds ratio	LCI	HCI	*p* value
HOTAIRCT	1.39	0.751	2.58	0.296
HOTAIRTT	1.27	0.419	3.87	0.672
LINC00951AG	0.939	0.528	1.67	0.831
LINC00951GG	0.195	0.0671	0.565	**0.004**
POLR2ECT	0.71	0.391	1.29	0.264
POLR2ECC	0.409	0.123	1.36	0.149
HULCAC	0.923	0.46	1.85	0.821
HULCCC	1.1	0.513	2.37	0.803

**Table 8 cancers-16-00537-t008:** Allele frequencies and genotype distributions demonstrating the association between POL2RE polymorphism and cancer risk prognostic factors in Esophageal Cancer (EC) and Esophageal adenocarcinoma (EAC) populations—Bold values denote statistically significant associations.

Genotype: POLR2E SNP: rs3787016, T>C (C/T)				
EC Population (N = 95)	Adenocarcinoma (EAC), N = 84 (%)	Squamous Cell Carcinoma, N = 6 (%)	OR (95% CI)	*p* Value
TT	41 (48.8)	1 (16.7)	1.00 (Ref.)	1.00 (Ref.)
CT	38 (45.2)	2 (33.3)	2.158 (0.1878–24.790)	0.6111
CC	5 (6)	3 (50)	**24.600 (2.130–284.17)**	**0.0105**
T allele	120 (71.4)	4 (33.4)	1.00 (Ref.)	1.00 (Ref.)
C allele	48 (28.6)	8 (66.6)	**5.000 (1.438–17.388)**	**0.0096**
EC population (N = 95)	Control Group, n = 121 (%)	Adenocarcinoma (EAC), N = 84 (%)	OR (95% CI)	*p* value
TT	43 (35.6)	41 (48.8)	1.00 (Ref.)	1.00 (Ref.)
CT	57 (47.1)	38 (45.2)	0.6992 (0.3864–1.265)	0.2912
CC	21 (17.3)	5 (6)	**0.2497 (0.08606–0.7246)**	**0.0114**
T allele	143 (59.2)	120 (71.4)	1.00 (Ref.)	1.00 (Ref.)
C allele	99 (40.8)	48 (28.6)	**0.5778 (0.3790–0.8808)**	**0.0119**
EAC subpopulation (N = 84/95)-Perineural Invasion (PNI) Status	Control Group, n = 121 (%)	Positive, N = 40 (%)	OR (95% CI)	*p* value
TT	43 (35.6)	23 (57.5)	1.00 (Ref.)	1.00 (Ref.)
CT	57 (47.1)	16 (40)	0.5248 (0.2476–1.112)	0.1299
CC	21 (17.3)	1 (2.5)	**0.08903 (0.01124–0.7052)**	**0.0051**
T allele	143 (59.2)	62 (77.5)	1.00 (Ref.)	1.00 (Ref.)
C allele	99 (40.8)	18 (22.5)	**0.4194 (0.2338–0.7521)**	**0.0031**
EAC subpopulation (N = 84/95)-Perineural Invasion (PNI) Status	Negative, N = 44 (%)	Positive, N = 40 (%)	OR (95% CI)	*p* value
TT	18 (40.9)	23 (57.5)	1.00 (Ref.)	1.00 (Ref.)
CT	22 (50)	16 (40)	0.5692 (0.2333–1.389)	0.2629
CC	4 (9.1)	1 (2.5)	0.1957 (0.02007–1.907)	0.1783
T allele	58 (65.9)	62 (77.5)	1.00 (Ref.)	1.00 (Ref.)
C allele	30 (34.1)	18 (22.5)	0.5613 (0.2828–1.114)	0.1238
EAC subpopulation (N = 84/95)-Lymphovascular Invasion (LVI) Status	Control Group, n = 121 (%)	Positive (N = 35) (%)	OR (95% CI)	*p* value
TT	43 (35.6)	20 (57.1)	1.00 (Ref.)	1.00 (Ref.)
CT	57 (47.1)	15 (42.9)	0.5658 (0.2599–1.232)	0.1713
CC	21 (17.3)	0 (0)	**0.04935 (0.002845–0.8559)**	**0.0022**
T allele	143 (59.2)	55 (78.6)	1.00 (Ref.)	1.00 (Ref.)
C allele	99 (40.8)	15 (21.4)	**0.3939 (0.2107–0.7367)**	**0.0030**
EAC subpopulation (N = 84/95)-Lymphovascular Invasion (LVI) Status	Negative, N = 49 (%)	Positive, N = 35 (%)	OR (95% CI)	*p* value
TT	21 (42.9)	20 (57.1)	1.00 (Ref.)	1.00 (Ref.)
CT	23 (46.9)	15 (42.9)	0.6848 (0.2802–1.674)	0.4980
CC	5 (10.2)	0 (0)	0.09534 (0.004949–1.837)	0.0593
T allele	65 (66.4)	55 (78.6)	1.00 (Ref.)	1.00 (Ref.)
C allele	33 (33.6)	15 (21.4)	0.5372 (0.2646–1.091)	0.1185
EAC subpopulation (N = 84/95)-Perivascular Invasion (PVI) Status	Control Group, n = 121 (%)	Positive, N = 39 (%)	OR (95% CI)	*p* value
TT	43 (35.6)	19 (48.7)	1.00 (Ref.)	1.00 (Ref.)
CT	57 (47.1)	18 (46.2)	0.7147 (0.3353–1.523)	0.4414
CC	21 (17.3)	2 (5.1)	**0.2155 (0.04584–1.013)**	**0.0477**
T allele	143 (59.2)	56 (71.8)	1.00 (Ref.)	1.00 (Ref.)
C allele	99 (40.8)	22 (28.2)	**0.5675 (0.3255–0.9894)**	**0.0454**
EAC subpopulation (N = 84/95)-Perivascular Invasion (PVI) Status	Negative, N = 45 (%)	Positive, N = 39 (%)	OR (95% CI)	*p* value
TT	22 (48.9)	19 (48.7)	1.00 (Ref.)	1.00 (Ref.)
CT	20 (44.4)	18 (46.2)	1.042 (0.4302–2.524)	1.0000
CC	3 (6.7)	2 (5.1)	0.7719 (0.1164–5.120)	1.0000
T allele	64 (71.1)	56 (71.8)	1.00 (Ref.)	1.00 (Ref.)
C allele	26 (28.9)	22 (28.2)	0.9670 (0.4940–1.893)	1.0000
EAC subpopulation (N = 84/95)-Differentiation Grade (G) Status	Control Group, n = 121 (%)	G2–G3, N = 80 (%)	OR (95% CI)	*p* value
TT	43 (35.6)	39 (48.8)	1.00 (Ref.)	1.00 (Ref.)
CT	57 (47.1)	36 (45)	0.6964 (0.3815–1.271)	0.2844
CC	21 (17.3)	5 (6.2)	**0.2625 (0.09027–0.7634)**	**0.0119**
T allele	143 (59.2)	114 (71.3)	1.00 (Ref.)	1.00 (Ref.)
C allele	99 (40.8)	46 (28.7)	**0.5828 (0.3800–0.8940)**	**0.0147**
EAC subpopulation (N = 84/95)-Differentiation Grade (G) Status	Control Group, n = 121 (%)	Gx–G1, N = 4 (%)	OR (95% CI)	*p* value
TT	43 (35.6)	2 (50)	1.00 (Ref.)	1.00 (Ref.)
CT	57 (47.1)	2 (50)	0.7544 (0.1021–5.574)	1.0000
CC	21 (17.3)	0 (0)	0.4047 (0.01858–8.811)	1.0000
T allele	143 (59.2)	6 (75)	1.00 (Ref.)	1.00 (Ref.)
C allele	99 (40.8)	2 (25)	0.4815 (0.09518–2.436)	0.4798
EAC subpopulation (N = 84/95)-Differentiation Grade (G) Status	Gx–G1, N = 4 (%)	G2–G3, N = 80 (%)	OR (95% CI)	*p* value
TT	2 (50)	39 (48.8)	1.00 (Ref.)	1.00 (Ref.)
CT	2 (50)	36 (45)	0.9231 (0.1234–6.904)	1.0000
CC	0 (0)	5 (6.2)	0.6962 (0.02938–16.500)	1.0000
T allele	6 (75)	114 (71.3)	1.00 (Ref.)	1.00 (Ref.)
C allele	2 (25)	46 (28.7)	1.211 (0.2355–6.221)	1.0000
EAC subpopulation (N = 84/95)-Differentiation Grade (G) Status	G2, N = 31 (%)	G3, N = 49 (%)	OR (95% CI)	*p* value
TT	11 (35.5)	28 (57.1)	1.00 (Ref.)	1.00 (Ref.)
CT	17 (54.8)	19 (38.8)	0.4391 (0.1687–1.143)	0.1009
CC	3 (9.7)	2 (4.1)	0.2619 (0.03837–1.788)	0.3065
T allele	39 (62.9)	75 (76.5)	1.00 (Ref.)	1.00 (Ref.)
C allele	23 (37.1)	23 (23.5)	0.5200 (0.2593–1.043)	0.0743
EAC subpopulation (N = 84/95)-Signet Ring Cell (SRC) Status	Control Group, n = 121 (%)	SRC positive, N = 20 (%)	OR (95% CI)	*p* value
TT	43 (35.6)	13 (65)	1.00 (Ref.)	1.00 (Ref.)
CT	57 (47.1)	5 (25)	**0.2901 (0.09610–0.8761)**	**0.0381**
CC	21 (17.3)	2 (10)	0.3150 (0.06502–1.526)	0.2080
T allele	143 (59.2)	31 (77.5)	1.00 (Ref.)	1.00 (Ref.)
C allele	99 (40.8)	9 (22.5)	**0.4194 (0.1912–0.9197)**	**0.0342**
EAC subpopulation (N = 84/95)-Lauren Classification Status	Control Group, n = 121 (%)	Diffuse Type positive, N = 15 (%)	OR (95% CI)	*p* value
TT	43 (35.6)	9 (60)	1.00 (Ref.)	1.00 (Ref.)
CT	57 (47.1)	6 (40)	0.5029 (0.1663–1.521)	0.2705
CC	21 (17.3)	0 (0)	0.106 (0.005912–1.918)	0.0519
T allele	143 (59.2)	24 (80)	1.00 (Ref.)	1.00 (Ref.)
C allele	99 (40.8)	6 (20)	**0.3611 (0.1424–0.9160)**	**0.0290**
EAC subpopulation (N = 84/95)-Lauren Classification Status	Intestinal Type positive, N = 28 (%)	Diffuse Type positive, N = 15 (%)	OR (95% CI)	*p* value
TT	11 (39.3)	9 (60)	1.00 (Ref.)	1.00 (Ref.)
CT	16 (57.1)	6 (40)	0.4583 (0.1265–1.661)	0.3357
CC	1 (3.6)	0 (0)	0.403 (0.01466–11.103)	1.0000
T allele	38 (67.85)	24 (80)	1.00 (Ref.)	1.00 (Ref.)
C allele	18 (32.15)	6 (20)	0.5278 (0.1836–1.517)	0.3148
EC-Preop Tumor Marker CEA	Control Group, n = 121 (%)	CEA positive, (>5 ng/mL), N = 19 (%)	OR (95% CI)	*p* value
TT	43 (35.6)	6 (31.6)	1.00 (Ref.)	1.00 (Ref.)
CT	57 (47.1)	13 (68.4)	1.635 (0.5746–4.650)	0.4492
CC	21 (17.3)	0 (0)	0.1556 (0.008367–2.895)	0.1684
T allele	143 (59.2)	25 (65.8)	1.00 (Ref.)	1.00 (Ref.)
C allele	99 (40.8)	13 (34.2)	0.7511 (0.3665–1.540)	0.4801
EC-Preop Tumor Marker CEA	CEA negative (<5 ng/mL), N = 59 (%)	CEA positive, (>5 ng/mL), N = 19 (%)	OR (95% CI)	*p* value
TT	30 (50.9)	6 (31.6)	1.00 (Ref.)	1.00 (Ref.)
CT	24 (40.6)	13 (68.4)	2.708 (0.8956–8.190)	0.1090
CC	5 (8.5)	0 (0)	0.4266 (0.02088–8.716)	1.0000
T allele	84 (71.2)	25 (65.8)	1.00 (Ref.)	1.00 (Ref.)
C allele	34 (28.8)	13 (34.2)	1.285 (0.5890–2.802)	0.5465
EC-Preop Tumor Marker Ca19.9	Control Group, n = 121 (%)	Ca19.9 positive, (>37 U/mL), N = 17 (%)	OR (95% CI)	*p* value
TT	43 (35.6)	10 (58.8)	1.00 (Ref.)	1.00 (Ref.)
CT	57 (47.1)	6 (35.3)	0.4526 (0.1526–1.342)	0.1809
CC	21 (17.3)	1 (5.9)	0.2048 (0.02454–1.708)	0.1587
T allele	143 (59.2)	26 (76.5)	1.00 (Ref.)	1.00 (Ref.)
C allele	99 (40.8)	8 (23.5)	0.4444 (0.1932–1.022)	0.0605
EC-Preop Tumor Marker Ca19.9	Ca19.9 negative (<37 U/mL), N = 59 (%)	Ca19.9 positive, (>37 U/mL), N = 17 (%)	OR (95% CI)	*p* value
TT	25 (42.4)	10 (58.8)	1.00 (Ref.)	1.00 (Ref.)
CT	30 (50.9)	6 (35.3)	0.5000 (0.1594–1.568)	0.2668
CC	4 (6.7)	1 (5.9)	0.6250 (0.06196–6.305)	1.0000
T allele	80 (67.9)	26 (76.5)	1.00 (Ref.)	1.00 (Ref.)
C allele	38 (32.1)	8 (23.5)	0.6478 (0.2682–1.564)	0.4004

**Table 9 cancers-16-00537-t009:** Allele frequencies and genotype distributions demonstrating the association between HULC polymorphism and cancer risk prognostic factors in Esophageal Cancer (EC) and Esophageal adenocarcinoma (EAC) populations.

Genotype: HULC SNP: rs7763881, A>C (C/A)				
EC Population (N = 95)	Adenocarcinoma (EAC), N = 84 (%)	Squamous Cell Carcinoma, N = 6 (%)	OR (95% CI)	*p* Value
AA	26 (31)	1 (16.7)	1.00 (Ref.)	1.00 (Ref.)
AC	37 (44)	4 (66.6)	2.811 (0.2967–26.629)	0.6411
CC	21 (25)	1 (16.7)	1.238 (0.07295–21.012)	1.0000
A allele	89 (53)	6 (50)	1.00 (Ref.)	1.00 (Ref.)
C allele	79 (47)	6 (50)	1.127 (0.3490–3.636)	1.0000
EAC subpopulation (N = 84/95)-Perineural Invasion (PNI) Status	Control Group, n = 121 (%)	Positive, N = 40 (%)	OR (95% CI)	*p* value
AA	35 (28.9)	12 (30)	1.00 (Ref.)	1.00 (Ref.)
AC	63 (52.1)	19 (47.5)	0.8796 (0.3825–2.023)	0.8315
CC	23 (19)	9 (22.5)	1.141 (0.4148–3.140)	0.8014
A allele	133 (55)	43 (53.7)	1.00 (Ref.)	1.00 (Ref.)
C allele	109 (45)	37 (46.3)	1.050 (0.6321–1.744)	0.8972
EAC subpopulation (N = 84/95)-Perineural Invasion (PNI) Status	Negative, N = 44 (%)	Positive, N = 40 (%)	OR (95% CI)	*p* value
AA	14 (31.8)	12 (30)	1.00 (Ref.)	1.00 (Ref.)
AC	18 (40.9)	19 (47.5)	1.231 (0.4506–3.365)	0.7994
CC	12 (27.3)	9 (22.5)	0.8750 (0.2747–2.787)	1.0000
A allele	46 (52.3)	43 (53.7)	1.00 (Ref.)	1.00 (Ref.)
C allele	42 (47.7)	37 (46.3)	0.9424 (0.5137–1.729)	0.8779
EAC subpopulation (N = 84/95)-Lymphovascular Invasion (LVI) Status	Control Group, n = 121 (%)	Positive, N = 35 (%)	OR (95% CI)	*p* value
AA	35 (28.9)	12 (34.3)	1.00 (Ref.)	1.00 (Ref.)
AC	63 (52.1)	14 (40)	0.6481 (0.2702–1.555)	0.3677
CC	23 (19)	9 (25.7)	1.141 (0.4148–3.140)	0.8014
A allele	133 (55)	38 (54.3)	1.00 (Ref.)	1.00 (Ref.)
C allele	109 (45)	32 (45.7)	1.028 (0.6023–1.753)	1.0000
EAC subpopulation (N = 84/95)-Lymphovascular Invasion (LVI) Status	Negative, N = 49 (%)	Positive, N = 35 (%)	OR (95% CI)	*p* value
AA	14 (28.6)	12 (34.3)	1.00 (Ref.)	1.00 (Ref.)
AC	23 (46.9)	14 (40)	0.7101 (0.2566–1.966)	0.6060
CC	12 (24.5)	9 (25.7)	0.8750 (0.2747–2.787)	1.0000
A allele	51 (52.1)	38 (54.3)	1.00 (Ref.)	1.00 (Ref.)
C allele	47 (47.9)	32 (45.7)	0.9138 (0.4940–1.690)	0.8755
EAC subpopulation (N = 84/95)-Perivascular Invasion (PVI) Status	Control Group, n = 121 (%)	Positive, N = 39 (%)	OR (95% CI)	*p* value
AA	35 (28.9)	15 (38.5)	1.00 (Ref.)	1.00 (Ref.)
AC	63 (52.1)	13 (33.3)	0.4815 (0.2058–1.127)	0.1245
CC	23 (19)	11 (28.2)	1.116 (0.4362–2.855)	0.8153
A allele	133 (55)	43 (55.2)	1.00 (Ref.)	1.00 (Ref.)
C allele	109 (45)	35 (44.8)	0.9932 (0.5945–1.659)	1.0000
EAC subpopulation (N = 84/95)-Perivascular Invasion (PVI) Status	Negative, N = 45 (%)	Positive, N = 39 (%)	OR (95% CI)	*p* value
AA	11 (24.4)	15 (38.5)	1.00 (Ref.)	1.00 (Ref.)
AC	24 (53.4)	13 (33.3)	0.3972 (0.1418–1.113)	0.1217
CC	10 (22.2)	11 (28.2)	0.8067 (0.2536–2.566)	0.7738
A allele	46 (51.1)	43 (55.2)	1.00 (Ref.)	1.00 (Ref.)
C allele	44 (48.9)	35 (44.8)	0.8510 (0.4631–1.564)	0.6438
EAC subpopulation (N = 84/95)-Differentiation Grade (G) Status	Control Group, n = 121 (%)	G2–G3, N = 80 (%)	OR (95% CI)	*p* value
AA	35 (28.9)	24 (30)	1.00 (Ref.)	1.00 (Ref.)
AC	63 (52.1)	36 (45)	0.8333 (0.4300–1.615)	0.6143
CC	23 (19)	20 (25)	1.268 (0.5737–2.803)	0.6859
A allele	133 (55)	84 (52.5)	1.00 (Ref.)	1.00 (Ref.)
C allele	109 (45)	76 (47.5)	1.104 (0.7396–1.648)	0.6828
EAC subpopulation (N = 84/95)-Differentiation Grade (G) Status	Control Group, n = 121 (%)	Gx–G1, N = 4 (%)	OR (95% CI)	*p* value
AA	35 (28.9)	2 (50)	1.00 (Ref.)	1.00 (Ref.)
AC	63 (52.1)	1 (25)	0.2778 (0.02430–3.175)	0.5524
CC	23 (19)	1 (25)	0.7609 (00.06513–8.888)	1.0000
A allele	133 (55)	5 (62.5)	1.00 (Ref.)	1.00 (Ref.)
C allele	109 (45)	3 (37.5)	0.7321 (0.1711–3.133)	0.7342
EAC subpopulation (N = 84/95)-Differentiation Grade (G) Status	Gx–G1, N = 4 (%)	G2–G3, N = 80 (%)	OR (95% CI)	*p* value
AA	2 (50)	24 (30)	1.00 (Ref.)	1.00 (Ref.)
AC	1 (25)	36 (45)	3.000 (0.2573–34.974)	0.5639
CC	1 (25)	20 (25)	1.667 (0.1405–19.770)	1.0000
A allele	5 (62.5)	84 (52.5)	1.00 (Ref.)	1.00 (Ref.)
C allele	3 (37.5)	76 (47.5)	1.508 (0.3485–6.525)	0.7239
EAC subpopulation (N = 84/95)-Differentiation Grade (G) Status	G2, N = 31 (%)	G3, N = 49 (%)	OR (95% CI)	*p* value
AA	7 (22.6)	17 (34.7)	1.00 (Ref.)	1.00 (Ref.)
AC	16 (51.6)	20 (40.8)	0.5147 (0.1715–1.545)	0.2852
CC	8 (25.8)	12 (24.5)	0.6176 (0.1760–2.167)	0.5316
A allele	30 (48.4)	54 (55.1)	1.00 (Ref.)	1.00 (Ref.)
C allele	32 (51.6)	44 (44.9)	0.7639 (0.4037–1.445)	0.4215
EAC subpopulation (N = 84/95)-Signet Ring Cell (SRC) Status	Control Group, n = 121 (%)	SRC positive, N = 20 (%)	OR (95% CI)	*p* value
AA	35 (28.9)	7 (35)	1.00 (Ref.)	1.00 (Ref.)
AC	63 (52.1)	7 (35)	0.5556 (0.1801–1.714)	0.3788
CC	23 (19)	6 (30)	1.304 (0.3886–4.378)	0.7588
A allele	133 (55)	21 (52.5)	1.00 (Ref.)	1.00 (Ref.)
C allele	109 (45)	20 (47.5)	1.162 (0.5989–2.255)	0.7352
EAC subpopulation (N = 84/95)-Lauren Classification Status	Control Group, n = 121 (%)	Diffuse Type positive, N = 15 (%)	OR (95% CI)	*p* value
AA	35 (28.9)	7 (46.7)	1.00 (Ref.)	1.00 (Ref.)
AC	63 (52.1)	5 (33.3)	0.3968 (0.1171–1.344)	0.2062
CC	23 (19)	3 (20)	0.6522 (0.1527–2.785)	0.7303
A allele	133 (55)	19 (63.35)	1.00 (Ref.)	1.00 (Ref.)
C allele	109 (45)	11 (36.65)	0.7064 (0.3223–1.548)	0.4391
EAC subpopulation (N = 84/95)-Lauren Classification Status	Intestinal Type positive, N = 28 (%)	Diffuse Type positive, N = 15 (%)	OR (95% CI)	*p* value
AA	10 (35.7)	7 (46.7)	1.00 (Ref.)	1.00 (Ref.)
AC	13 (46.4)	5 (33.3)	0.5495 (0.1337–2.258)	0.4887
CC	5 (17.9)	3 (20)	0.8571 (0.1524–4.821)	1.0000
A allele	33 (58.9)	19 (63.35)	1.00 (Ref.)	1.00 (Ref.)
C allele	23 (41.1)	11 (36.65)	0.8307 (0.3331–2.072)	0.8178
EC-Preop Tumor Marker CEA	Control Group, n = 121 (%)	CEA positive, (>5 ng/mL), N = 19 (%)	OR (95% CI)	*p* value
AA	35 (28.9)	4 (21.1)	1.00 (Ref.)	1.00 (Ref.)
AC	63 (52.1)	11 (57.8)	1.528 (0.4524–5.159)	0.5722
CC	23 (19)	4 (21.1)	1.522 (0.3454–6.703)	0.7068
A allele	133 (55)	19 (50)	1.00 (Ref.)	1.00 (Ref.)
C allele	109 (45)	19 (50)	1.220 (0.6153–2.420)	0.6023
EC-Preop Tumor Marker CEA	CEA negative (<5 ng/mL), N = 59 (%)	CEA positive, (>5 ng/mL), N = 19 (%)	OR (95% CI)	*p* value
AA	20 (33.9)	4 (21.1)	1.00 (Ref.)	1.00 (Ref.)
AC	22 (37.3)	11 (57.8)	2.500 (0.6847–9.128)	0.2260
CC	17 (28.8)	4 (21.1)	1.176 (0.2548–5.431)	1.0000
A allele	62 (52.6)	19 (50)	1.00 (Ref.)	1.00 (Ref.)
C allele	56 (47.4)	19 (50)	1.107 (0.5327–2.301)	0.8528
EC-Preop Tumor Marker Ca19.9	Control Group, n = 121 (%)	Ca19.9 positive, (>37 U/mL), N = 17 (%)	OR (95% CI)	*p* value
AA	35 (28.9)	7 (41.2)	1.00 (Ref.)	1.00 (Ref.)
AC	63 (52.1)	6 (35.3)	0.4762 (0.1483–1.529)	0.2335
CC	23 (19)	4 (23.5)	0.8696 (0.2284–3.310)	1.0000
A allele	133 (55)	20 (58.9)	1.00 (Ref.)	1.00 (Ref.)
C allele	109 (45)	14 (41.1)	0.8541 (0.4122–1.770)	0.7157
EC-Preop Tumor Marker Ca19.9	Ca19.9 negative (<37 U/mL), N = 59 (%)	Ca19.9 positive, (>37 U/mL), N = 17 (%)	OR (95% CI)	*p* value
AA	17 (28.8)	7 (41.2)	1.00 (Ref.)	1.00 (Ref.)
AC	25 (42.4)	6 (35.3)	0.5829 (0.1665–2.040)	0.5251
CC	17 (28.8)	4 (23.5)	0.5714 (0.1408–2.319)	0.5030
A allele	59 (50)	20 (58.9)	1.00 (Ref.)	1.00 (Ref.)
C allele	59 (50)	14 (41.1)	0.7000 (0.3233–1.516)	0.4373

## Data Availability

Data contained within the article.

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
