# Peer review of "Genetic Impact of HOTAIR, LINC00951, POLR2E and HULC Polymorphisms in Histopathological and Laboratory Prognostic Factors in Esophageal Cancer in the West: A Case-Control Study"

_cancers, 2024, doi:10.3390/cancers16030537_

Round 1
Reviewer 1 Report
Comments and Suggestions for Authors
The manuscript describes the prognostic analysis of 4 polymorphisms of the lncRNAs HOTAIR, LINC00951, POLR2E and HULC in esophageal cancer (EC). The study is of interest to scientists in the field. There are nonetheless several limitations to the analysis (some mentioned in discussion) that would merit testing.
The number of samples is very low in certain experimental groups (n < 20), which makes the analyses not robust enough. The number of patients analyzed needs to be increased.
Samples and controls are not of the same origin (CE vs blood), which may also limits analysis. Is it possible to investigate the SNP status on FFPE in the CE and healthy part ? or on blood samples from EC patients ?
Most patients are men (> 90%) which is surprising. Is there a sex biais in EC in Greece ? Is it the same in other ethinicities and could it be a confounding factor in the analysis ?
There is no justification for studying the 4 SNPs and not other SNPS. What is the rationale for selecting these 4 SNPs for further analysis.
I would thus suggest "Major revision", and the english is fine, even if the abstract and simple summary sections might need some rephrasing.
The manuscript describes the prognostic analysis of 4 polymorphisms of the lncRNAs HOTAIR, LINC00951, POLR2E and HULC in esophageal cancer (EC). The study is of interest to scientists in the field. There are nonetheless several limitations to the analysis (some mentioned in discussion) that would merit testing.
The number of samples is very low in certain experimental groups (n < 20), which makes the analyses not robust enough. The number of patients analyzed needs to be increased.
Samples and controls are not of the same origin (CE vs blood), which may also limits analysis. Is it possible to investigate the SNP status on FFPE in the CE and healthy part ? or on blood samples from EC patients ?
Most patients are men (> 90%) which is surprising. Is there a sex biais in EC in Greece ? Is it the same in other ethinicities and could it be a confounding factor in the analysis ?
There is no justification for studying the 4 SNPs and not other SNPS. What is the rationale for selecting these 4 SNPs for further analysis.
I would thus suggest "Major revision", and the english is fine, even if the abstract and simple summary sections might need some rephrasing.
Author Response
"Please see the attachment."

Reviewer 2 Report
Comments and Suggestions for Authors
1. Introduction. Paragraph about histopathological types should be shorten, but additional information about molecular background should be added.
2. Does the consent of the Bioethics Comittee also include the control group?
3. It is necessary to add information about PCR and RFLP conditions, also the name of enzymes, sequences of the primers. The same for AS-PCR. In cited publication number 24(2022 Oct 6;58(10):1399. doi: 10.3390/medicina58101399.)Makrantonakis, A.-E.; Zografos, E.; Gazouli, M.; Dimitrakakis, K.; Toutouzas, K.G.; Zografos, C.G.; Kalapanida, D.; Tsiakou, 644 A.; Samelis, G.; Zagouri, F. PD-L1 Gene Polymorphisms rs822336 G>C and rs822337 T>A: Promising Prognostic 645 Markers in Triple Negative Breast Cancer Patients. Medicina 2022, 58, 1399. there is description of PCR for different SNPs and no information about AS-PCR
4. Information about the number of cases in control and study groups should be placed not in results section
5. Has compliance of genotypes and allels distribution for all analysed SNPs with H-W equilibrum been checked?
Results. In this section the most important results should be described in text, not only placed in tables
6. Paragraph from line 220 to 230 should be also put into the table
7. Table 1 is very long, it could be shortened and condensed
8. Tables 2 and 3 should be combined and moved to supplementary materials.
9. Fragments of table 4 could be show as graphs/figures. The same for tables 5, 7 and 9. In present form it is difficult to read.
10. Lack of short results summary at the end of this part of manuscript.
11. Discussion is too general. Paragraph from line 425 to 434 should be transfer to introduction. The most important findings should be underlined during discussion.
Author Response
"Please see the attachment."

Round 2
Reviewer 1 Report
Comments and Suggestions for Authors
I scrolled throught the comments of the authors and the alterations in the manuscript and I found the newest version clearly improved and deemed publishable in its later form.
Reviewer 2 Report
Comments and Suggestions for Authors
The manuscripts is properly corrected and modified.